**Probabilistic Hierarchical Interpolation and Interpretable Neural Network Configurations for**
**Flood Prediction**
Mostafa Saberian1, Vidya Samadi2,3*, Ioana Popescu4
1. The Glenn Department of Civil Engineering, Clemson University, Clemson, SC
2. Department of Agricultural Sciences, Clemson University, Clemson, SC.
3. Artificial Intelligence Research Institute for Science and Engineering (AIRISE), Clemson
University, Clemson, SC, USA
4. Department of Hydroinformatics and Socio-Technical Innovation, IHE Delft Institute for Water
Education, Delft, the Netherlands
*Corresponding author: samadi@clemson.edu
**Abstract**
The past few years have witnessed the rise of neural networks (NNs) applications for hydrological time
series modeling. By virtue of their capabilities, NN models can achieve unprecedented levels of
performance when learning how to solve increasingly complex rainfall-runoff processes via data, making
them pivotal for the development of computational hydrologic tasks such as flood predictions. The NN
models should, to be considered practical, provide a probabilistic understanding of the model mechanisms
and predictions and hints on what could perturb the model. In this paper, we developed two NN models,
i.e., Neural Hierarchical Interpolation for Time Series Forecasting (N-HiTS) and Network-Based
Expansion Analysis for Interpretable Time Series Forecasting (N-BEATS) with a probabilistic multi-
quantile objective and benchmarked them with long short-term memory (LSTM) for flood prediction across
two headwater streams in Georgia and North Carolina, USA. To generate a probabilistic prediction, a Multi-
Quantile Loss was used to assess the 95th percentile prediction uncertainty (95PPU) of multiple flooding
events. Extensive experiments demonstrated the advantages of hierarchical interpolation and interpretable
architecture, where both N-HiTS and N-BEATS provided an average accuracy improvement of ~5% over
the LSTM benchmarking model. On a variety of flooding events, both N-HiTS and N-BEATS demonstrated
significant performance improvements over the LSTM benchmark and showcased their probabilistic
predictions by specifying a likelihood objective.
**Keywords:** Probabilistic Flood Prediction; Neural Networks; N-HiTS; N-BEATS; LSTM; Headwater
Stream.
**Key Points**
•   N-HiTS and N-BEATS predictions reflect interpretability and hierarchical representations of data
to reduce neural network complexities.
•   Both N-HiTS and N-BEATS models outperformed the LSTM in mathematically defining
uncertainty bands.
•   Predicting the magnitude of the recession curve of flood hydrographs was particularly challenging
for all models.

**Plain Language Summary**

Recent progress in NN accelerated improvements in the performance of catchment modeling. Yet flood modeling remains a very difficult task. Focusing on two headwater streams, we developed N-HiTS and N-BEATS models and benchmarked them with LSTM to predict flooding. N-HiTS and N-BEATS outperformed LSTM for flood predictions. We demonstrated how the proposed models can be augmented with an uncertainty approach to predict flooding that is interpretable without considerable loss in accuracy.

**1. Introduction**

The past few years have witnessed a rapid surge in the neural networks (NN) applications in hydrology. As these opaque, data-driven models are increasingly employed for critical hydrological predictions, the hydrology community has placed growing emphasis on developing trustworthy and interpretable NN models. However, maintaining coherence while producing accurate predictions can be a challenging problem (Olivares et al., 2024). There is a general agreement on the importance of providing probabilistic NN prediction (Sadeghi Tabas and Samadi, 2022), especially in the case of flood prediction (Martinaitis et al., 2023).

Flood occurrences have witnessed an alarming surge in frequency and severity globally. Jonkman (2005) studied a natural disaster database (EM-DAT, 2023) and reported that over 27 years, more than 175000 people died, and close to 2.2 billion were affected directly by floods worldwide. These numbers are likely an underestimation due to unreported events (Nevo et al., 2022). In addition, the United Nations Office for Disaster Risk Reduction reported that flooding has been the most frequent, widespread weather-related natural disaster since 1995, claiming over 600,000 lives, affecting around 4 billion people globally, and causing annual economic damage of more than 100 billion USD (UNISDR, 2015). This escalating trend has necessitated the need for better flood prediction and management strategies. Scholars have successfully implemented different flood models such as deterministic (e.g., Roelvink et al., 2009, Thompson and Frazier, 2014; Barnard et al., 2014; Erikson et al., 2018) and physically based flood models (e.g., Basso et al., 2016; Chen et al., 2016; Pourreza-Bilondi et al., 2017; Saksena et al., 2019; Refsgaard et al., 2021) in various environmental systems over the past several decades. These studies have heightened the need for

precise flood prediction (Samadi et al., 2025), they have also unveiled limitations inherent in existing
deterministic and physics-based models.
While evidence suggests that both deterministic and physics-based approaches are meaningful and useful
(Sukovich et al., 2014; Zafarmomen et al., 2024), their forecasts rest heavily on imprecise and subjective
expert opinion; there is a challenge for setting robust evidence-based thresholds to issue flood warnings and
alerts (Palmer, 2012). Moreover, many of these traditional flood models, particularly physically explicit
models, rely too strongly on a particular choice of numerical approximation and describe multiple process
parameterizations only within a fixed spatial architecture (e.g., Clark et al., 2015). Recent NN models have
shown promising results across a large variety of flood modeling applications (e.g., Nevo et al., 2022; Pally
and Samadi, 2022; Dasgupta et al., 2023; Zhang et al., 2023; Zafarmomen and Samadi, 2025; Saberian et
al., 2025) and encourage the use of such methodologies as core drivers for neural flood prediction
(Windheuser et al., 2023).
Earlier adaptations of these intelligent techniques showed promising for flood prediction (e.g., Hsu et al.,
1995; Tiwari and Chatterjee, 2010). However, recent efforts have taken NN application to the next level,
providing uncertainty assessment (Sadeghi Tabas and Samadi, 2022) and improvements over various
spatio-temporal scales, regions, and processes (e.g., Kratzert et al., 2018; Park and Lee, 2023; Zhang et al.,
2023). Nevo et al., (2022) were the first scholars who employed long short-term memory (LSTM) for flood
stage prediction and inundation mapping, achieving notable success during the 2021 monsoon season. Soon
after, Russo et al. (2023) evaluated various NN models for predicting depth flood in urban systems,
highlighting the potential of data-driven models for urban flood prediction. Similarly, Defontaine et al.
(2023) emphasized the role of NN algorithms in enhancing the reliability of flood predictions, particularly
in the context of limited data availability. Windheuser et al., (2023) studied flood gauge height forecasting
using images and time series data for two gauging stations in Georgia, USA. They used multiple NN models
such as Convolutional Neural Network (ConvNet/CNN) and LSTM to forecast floods in near real-time (up
to 72 hours).
In a sequence, Wee et al., (2023) used Impact-Based Forecasting (IBF) to propose a Flood Impact-Based
Forecasting system (FIBF) using flexible fuzzy inference techniques, aiding decision-makers in a timely
response. Zou et al. (2023) proposed a Residual LSTM (ResLSTM) model to enhance and address flood
prediction gradient issues. They integrated Deep Autoregressive Recurrent (DeepAR) with four recurrent
neural networks (RNNs), including ResLSTM, LSTM, Gated Recurrent Unit (GRU), and Time
Feedforward Connections Single Gate Recurrent Unit (TFC-SGRU). They showed that ResLSTM achieved
superior accuracy. While these studies reported the superiority of NN models for flood modeling, they
highlighted a number of challenges, notably (i) the limited capability of proposed NN models to capture
the spatial variability and magnitudes of extreme data over time, (ii) the lack of a sophisticated mechanism
to capture different flood magnitudes and synthesize the prediction, and (iii) inability of the NN models to
process data in parallel and capture the relationships between all elements in a sequential manner.
Recent advances in neural time series forecasting showed promising results that can be used to address the
above challenges for flood prediction. Recent techniques include the adoption of the attention mechanism
and Transformer-inspired approaches (Fan et al. 2019; Alaa and van der Schaar 2019; Lim et al. 2021)
along with attention-free architectures composed of deep stacks of fully connected layers (Oreshkin et al.
2020).

All these approaches are relatively easy to scale up in terms of flood magnitudes (small to major flood
predictions), compared to LSTM and have proven to be capable of capturing spatiotemporal dependencies
(Challu et al., 2022). In addition, these architectures can capture input-output relationships implicitly while
they tend to be more computationally efficient. Many state-of-the-art NN approaches for flood forecasting
have been established based on LSTM. There are cell states in the LSTM networks that can be interpreted
as storage capacity often used in flood generation schemes. In LSTM, the updating of internal cell states
(or storages) is regulated through several gates: the first gate regulates the storage depletion, the second one
regulates storage fluctuations, and the third gate regulates the storages outflow (Tabas and Samadi, 2022).
The elaborate gated design of the LSTM partly solves the long-term dependency problem in flood time
series prediction (Fang et al., 2020), although, the structure of LSTMs is designed in a sequential manner
that cannot directly connect two nonadjacent portions (positions) of a time series.
In this paper, we developed attention-free architecture, i.e. Neural Hierarchical Interpolation for Time
Series Forecasting (N-HiTS; Challu et al., 2022) and Network-Based Expansion Analysis for Interpretable
Time Series Forecasting (N-BEATS; Oreshkin et al., 2020) and benchmarked these models with LSTM for
flood prediction. We developed fully connected N-BEATS and N-HiTS architectures using multi-rate data
sampling, synthesizing the flood prediction outputs via multi-scale interpolation.
We implemented all algorithms for flood prediction on two headwater streams i.e., the Lower Dog River,
Georgia, and the Upper Dutchmans Creek, North Carolina, USA to ensure that the results are reliable and
comparable. The results of N-BEATS and N-HiTS techniques were compared with the benchmarking
LSTM to understand how these techniques can improve the representations of rainfall and runoff
dispensing over a recurrence process. Notably, this study represents a pioneering effort, as to the best of
our knowledge, this is the first instance in which the application of N-BEATS and N-HiTS algorithms in
the field of flood prediction has been explored. The scope of this research will focus on:
**(i)    Flood prediction in a hierarchical fashion with interpretable outputs:** We built N-BEATS and
N-HiTS for flood prediction with a very deep stack of fully connected layers to implicitly capture input-
output relationships with hierarchical interpolation capabilities. The predictions also involve programming
the algorithms with decreasing complexity and aligning their time scale with the final output through multi-
scale hierarchical interpolation and interpretable architecture. Predictions were aggregated in a hierarchical
fashion that enabled the building of a very deep neural network with interpretable configurations.
**(ii)    Uncertainty quantification of the models by employing probabilistic approaches:** a Multi-
Quantile Loss (MQL) was used to assess the 95th percentile prediction uncertainty (95PPU) of multiple
flooding events. MQL was integrated as the loss function to account for probabilistic prediction. MQL
trains the model to produce probabilistic forecasts by predicting multiple quantiles of the distribution of
future values.
**(iii)    Exploring headwater stream response to flooding:** Understanding the dynamic response of
headwater streams to flooding is essential for managing downstream flood risks. Headwater streams
constitute the uppermost sections of stream networks, usually comprising 60% to 80% of a catchment area.
Given this substantial coverage and the tendency for precipitation to increase with elevation, headwater
streams are responsible for generating and controlling the majority of runoff in downstream portions
(MacDonald and Coe, 2007).
The remainder of this paper is structured as follows. Section 2 presents the case study and data, NN models,
performance metrics, and sensitivity and uncertainty approaches. Section 3 focuses on the results of flood
predictions including sensitivity and uncertainty assessment and computation efficiency. Finally, Section 4
concludes the paper.

**2. Methodology**
**2.1. Case Study and Data**
This research used two headwater gauging stations located at the Lower Dog River watershed, Georgia
(GA; USGS02337410, Dog River gauging station), and the Upper Dutchmans Creek watershed, North
Carolina (NC; USGS0214269560, Killian Creek gauging station). As depicted in Figures 1, the Lower Dog
River and the Upper Dutchmans Creek watersheds are in the west and north parts of two metropolitan cities,
Atlanta and Charlotte. The Lower Dog River stream gauge is established southeast of Villa Rica in Carroll
County, where the USGS has regularly monitored discharge data since 2007 in 15-minute increments. The
Lower Dog River is a stream with a length of 15.7 miles (25.3 km; obtained from the U.S. Geological
Survey [USGS] National Hydrography Dataset high-resolution flowline data), an average elevation of

851.94 meters, and the watershed area above this gauging station is 66.5 square miles (172 km2; obtained from the Georgia Department of Natural Resources). This watershed is covered by 15.2% residential area, 14.6% agricultural land, and ~70% forest (Munn et al., 2020).

Killian Creek gauging station at the Upper Dutchmans Creek watershed is established in Montgomery County, NC, where the USGS has regularly monitored discharge data since 1995 in 15-minute increments. The Upper Dutchmans Creek is a stream with a length of 4.9 miles (7.9 km), an average elevation of 642.2 meters (see Table 1), and the watershed area above this gauging station is 4 square miles (10.3 km2) with less than 3% residential area and about 93% forested land use (the United States Environmental Protection Agency).

The Lower Dog River has experienced significant flooding in the last decades. For example, in September 2009, the creek, along with most of northern GA, experienced heavy rainfall (5 inches, equal to 94 mm). The Lower Dog River, overwhelmed by large amounts of overland flow from saturated ground in the watershed, experienced massive flooding in September 2009 (Gotvald, 2010). The river crested at 33.8 feet (10.3 m) with a peak discharge of 59,900 cfs (1,700 m3/s), nearly six times the 100-year flood level (McCallum and Gotvald, 2010). In addition, Dutchmans Creek experienced significant flooding in February 2020. According to local news (WCCB Charlotte, 2020), the flood in Gaston County caused significant infrastructure damage and community disruption. Key impacts included the threatened collapse of the Dutchman's Creek bridge in Mt. Holly and the closure of Highway 7 in McAdenville, GA.

Table 1. The Lower Dog River and Upper Dutchmans Creek's physical characteristics.

| Watershed | USGS Station ID Number | Average Elevation (m) | Stream Length (km) | Watershed area (km2) |
|---|---|---|---|---|
| Lower Dog River watershed, GA | USGS02337410 | 851.9 | 25.3 | 172 |
| Upper Dutchmans Creek watershed, NC | USGS0214269560 | 642.2 | 7.9 | 10.3 |

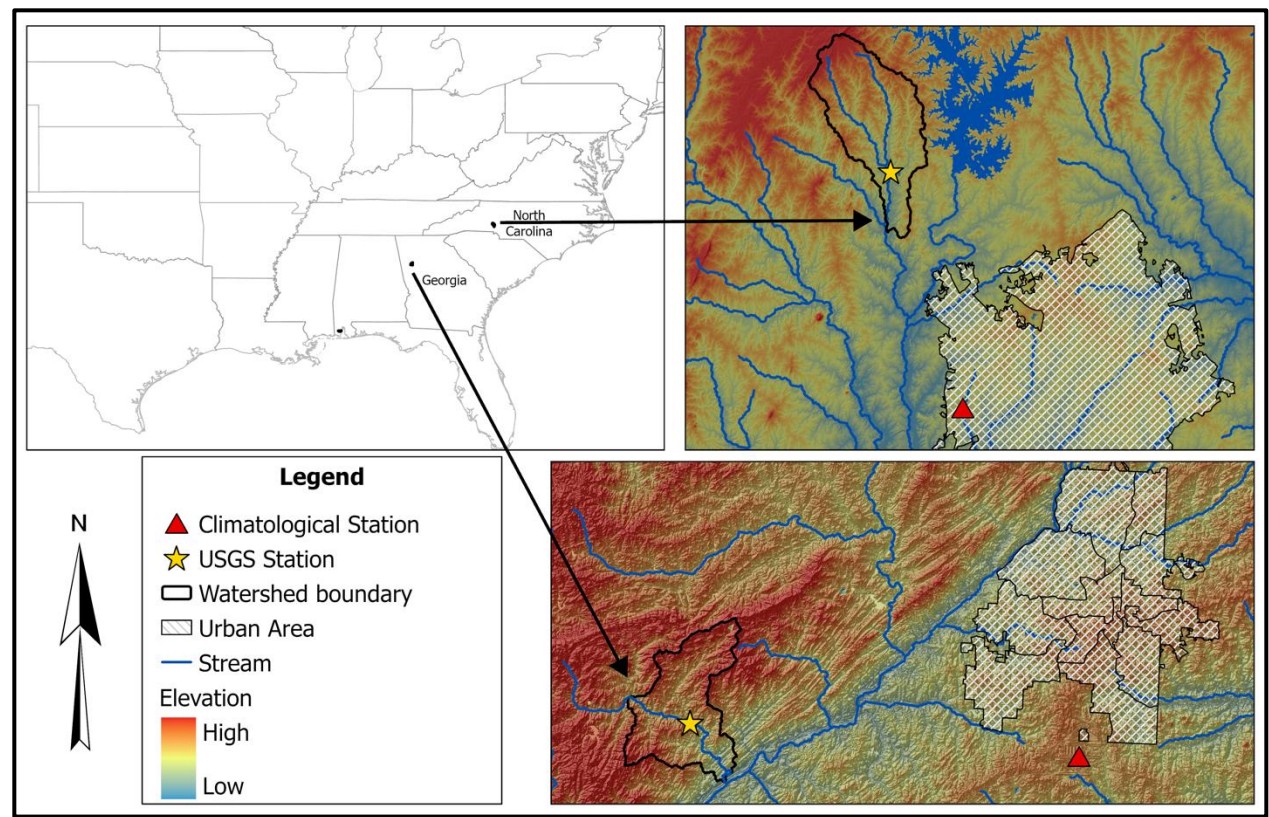

Figure 1. The Lower Dog River and The Upper Dutchmans Creek watersheds are in GA and NC. The proximity of the watersheds to Atlanta and Charlotte (urban area) are also displayed on the map.

To provide the meteorological forcing data, i.e., precipitation, temperature, and humidity, were extracted from the National Oceanic and Atmospheric Administration's (NOAA) Local Climatological Data (LCD). We used the NOAA precipitation, temperature, and humidity data of Atlanta Hartsfield Jackson International Airport and Charlotte Douglas Airport stations as an input for neural network algorithms. The data has been monitored since January 1, 1948, and July 22, 1941, with an hourly interval which was used as an input variable for constructing neural networks.

To fill in the missing values in the data, we used the spline interpolation method. We applied this method to fill the gaps in time series data, although the missing values were insignificant (less than 1%). In addition, we employed the Minimum Inter-Event Time (MIT) approach to precisely identify and separate individual storm events. The MIT-based event delineation is pivotal for accurately defining storm events. This method allowed us to isolate discrete rainfall episodes, aiding a comprehensive analysis of storm events. Moreover, it provided a basis for event-specific examination of flood responses, such as initial condition and cessation (loss), runoff generation, and runoff dynamics.

The hourly rainfall dataset consists of distinct rainfall occurrences, some consecutive and others clustered
with brief intervals of zero rainfall. As these zero intervals extend, we aim to categorize them into distinct
events. It's worth noting that even within a single storm event, we often encounter short periods of no
rainfall, known as intra-storm zero values. In the MIT method, we defined a storm event as a discrete rainfall
episode surrounded by dry periods both preceding and following it, determined by an MIT (Asquith et al.,
2005; Safaei-Moghadam et al., 2023).
There are many ways to determine MIT value. One practical approximation is using serial autocorrelation
between rainfall occurrences. MIT approach uses autocorrelation that measures the statistical dependency
of rainfall data at one point in time with data at earlier, or lagged times within the time series. The lag time
represents the gap between data points being correlated. When the lag time is zero, the autocorrelation
coefficient is unity, indicating a one-to-one correlation. As the lag time increases, the statistical correlation
diminishes, converging to a minimum value. This signifies the fact that rainfall events become
progressively less statistically dependent or, in other words, temporally unrelated. To pinpoint the optimal
MIT, we analyzed the autocorrelation coefficients for various lag times, observing the point at which the
coefficient approaches zero. This lag time signifies the minimum interval of no rainfall, effectively
delineating distinct rainfall events.

**2.2. NN Algorithms**
In this study, three distinct neural network (NN) architectures were developed to perform multi-horizon
flood forecasting. Each NN was coupled with a MQL objective to generate probabilistic predictions and
quantify predictive uncertainty. Throughout the manuscript, the term parameters are used exclusively to
refer to the network's weights and biases for clarity and consistency.

**2.2.1. LSTM**
LSTM is an RNN architecture widely used as a benchmark model for flood neural time series
modeling. LSTM networks are capable of selectively learning order dependence in sequence prediction
problems (Sadeghi Tabas and Samadi, 2022). These networks are powerful because they can capture the
temporal features, especially the long-term dependencies (Hochreiter et al., 2001) and are independent of
the length of the data sequences input, meaning that each sample is independent from another one.
The memory cell state within LSTM plays a crucial role in capturing extended patterns in data, making it
well-suited for dynamic time series modeling such as flood prediction. An LSTM cell uses the following
functions to compute flood prediction.

$$i_t = \sigma(A_i x_t + B_i h_{t-1} + c_i) \qquad \text{(Equation 1)}$$

$$f_t = \sigma(A_f x_t + B_f h_{t-1} + c_f) \qquad \text{(Equation 2)}$$

$$o_t = \sigma(A_o x_t + B_o h_{t-1} + c_o) \qquad \text{(Equation 3)}$$

$$m_t = f_t \odot m_{t-1} + i_t \odot tanh(A_g x_t + B_g h_{t-1} + c_g) \qquad \text{(Equation 4)}$$

$$h_t = o_t \odot tanh(m_t) \qquad \text{(Equation 5)}$$

Where $x_t$ and $h_t$ represent the input and the hidden state at time step $t$, respectively. $\odot$ denotes element-
wise multiplication, $tanh$ stands for the hyperbolic tangent activation function, and $\sigma$ represents the
sigmoid activation function. $A$, $B$, and $c$ are trainable weights and biases that undergo optimization during
the training process. $m_t$ and $h_t$ are cell states at time step $t$ that are employed in the input processing for
the next time step. $m_t$ represents the memory state responsible for preserving long-term information, while
$h_t$ represents the memory state preserving short-term information. The LSTM cell consists of a forget gate
$f_t$, an input gate $i_t$ and an output gate $o_t$ and has a cell state $m_t$. At every time step $t$, the cell gets the data
point $x_t$ with the output of the previous cell $h_{t-1}$ (Windheuser et al., 2023). The forget gate then defines if
the information is removed from the cell state, while the input gate evaluates if the information should be
added to the cell state and the output gate specifies which information from the cell state can be used for
the next cells.
We used two LSTM layers with 128 cells in the first two hidden layers as encoder layers, which were then
connected to two multilayer perceptron (MLP) layers with 128 neurons as decoder layers. The LSTM
simulation was performed with these input layers along with the *Adam* optimizer (Kingma and Ba,
2014), *tanh* activation function, and a single lagged dependent-variable value to train with a learning rate
of 0.001. The architecture of the proposed LSTM model is illustrated in Figure 2.

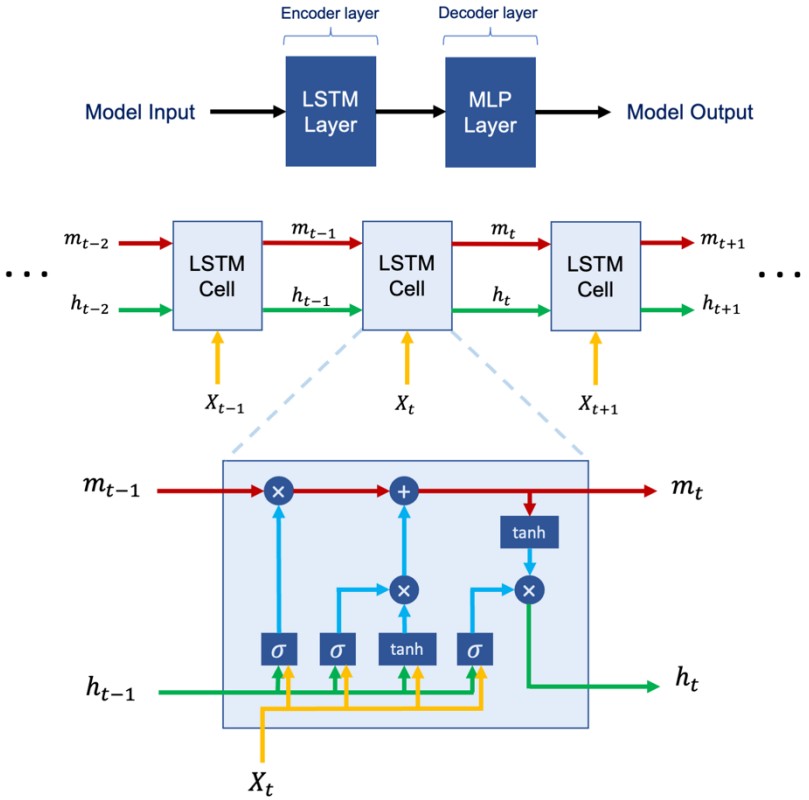


Figure 2. The structure of LSTM programmed in this research. We used *tanh* and *sigmoid* as activation
functions along with 2 layers of LSTM, 2 layers of MLP, and 128 cells in each layer.

### 2.2.2. N-BEATS

N-BEATS is a deep learning architecture based on backward and forward residual links and the very deep
stack of fully connected layers specifically designed for sequential data forecasting tasks (Oreshkin et al.,
2020). This architecture has several desirable properties including interpretability. The N-BEATS
architecture distinguishes itself from existing architecture in several ways. First, the algorithm approaches
forecasting as a non-linear multivariate regression problem instead of a sequence-to-sequence
challenge. Indeed, the core component of this architecture (as depicted in Figure 3) is a fully connected
non-linear regressor, which takes the historical data from a time series as input and generates multiple data
points for the forecasting horizon. Second, most existing time series architectures are quite limited in depth,
typically consisting of one to five LSTM layers. N-BEATS employs the residual principle to stack a
substantial number of layers together, as illustrated in Figure 3. In this configuration, the basic block not
only predicts the next output but also assesses its contribution to decomposing the input, a concept that is
referred to as "backcast" (see Oreshkin et al. 2020).
The basic building block in the architecture features a fork-like structure, as illustrated in Figure 3 (bottom).
The $l$-th block (for the sake of brevity, the block index $l$ is omitted from Figure 3) takes its respective input,
$x_l$, and produces two output vectors: $\hat{x}_l$ and $\hat{y}_l$. In the initial block of the model, $x_l$ corresponds to the
overall model input, which is a historical lookback window of a specific length, culminating with the most
recent observed data point. For the subsequent blocks, $x_l$ is derived from the residual outputs of the
preceding blocks. Each block generates two distinct outputs: 1. $\hat{y}_l$: This represents the forward forecast of
the block, spanning a duration of $H$ time units. 2. $\hat{x}_l$: This signifies the block's optimal estimation of $x_l$,
which is referred to "backcast." This estimation is made within the constraints of the functional space
available to the block for approximating signals (Oreshkin et al., 2020).
Internally, the fundamental building block is composed of two elements. The initial element involves a
fully connected network, which generates forward expansion coefficient predictors, $\theta_l^f$, and a backward
expansion coefficient predictor, $\theta_l^b$. The second element encompasses both backward basis layers, $g_l^b$, and
forward basis layers, $g_l^f$. These layers take the corresponding forward $\theta_l^f$ and backward $\theta_l^b$ expansion
coefficients as input, conduct internal transformations using a set of basis functions, and ultimately yield
the backcast, $\hat{x}_l$, and the forecast outputs, $\hat{y}_l$, as previously described by Oreshkin et al. (2020). The
following equations describe the first element:

$$h_{l,1} = FC_{l,1}(x_l), \quad h_{l,2} = FC_{l,2}(h_{l,1}), \quad h_{l,3} = FC_{l,3}(h_{l,2}), \quad h_{l,4} = FC_{l,4}(h_{l,3}). \qquad \text{(Equation 6)}$$

$$\theta_l^b = \text{LINEAR}_l^b(h_{l,4}), \qquad \theta_l^b = \text{LINEAR}_l^b(h_{l,4}) \qquad \text{(Equation 7)}$$

The LINEAR layer, in essence, functions as a straightforward linear projection, meaning $\theta_l^f = W_l^f h_{l,4}$. As
for the fully connected (FC) layer, it takes on the role of a conventional FC layer, incorporating RELU non-
linearity as an activation function.
The second element performs the mapping of expansion coefficients $\theta_l^f$ and $\theta_l^b$ to produce outputs using
basis layers, resulting in $\hat{y}_l = g_l^f(\theta_l^f)$ and $\hat{x}_l = g_l^b(\theta_l^b)$. This process is defined by the following equation:

$$\hat{y}_l = \sum_{i=1}^{\dim(\theta_l^f)} \theta_{l,i}^f v_i^f, \qquad \hat{x}_l = \sum_{i=1}^{\dim(\theta_l^b)} \theta_{l,i}^b v_i^b \qquad \text{(Equation 8)}$$

Within this context, $v_i^f$ and $v_i^b$ represent the basis vectors for forecasting and backcasting, respectively,
while $\theta_{l,i}^f$ corresponds to the $i$-th element of $\theta_l^f$.
The N-BEATS uses a novel hierarchical doubly residual architecture which is illustrated in Figure 3 (top
and middle). This framework incorporates two residual branches, one traversing the backcast predictions
of each layer, while the other traverses the forecast branch of each layer. The following equation describes
this process:

$$x_l = x_{l-1} - \hat{x}_{l-1} \quad , \quad \hat{y} = \sum_l \hat{y}_l \qquad \text{(Equation 9)}$$

As mentioned earlier, in the specific scenario of the initial block, its input corresponds to the model-level
input $x$. In contrast, for all subsequent blocks, the backcast residual branch $x_l$ can be conceptualized as
conducting a sequential analysis of the input signal. The preceding block eliminates the portion of the signal
$\hat{x}_{l-1}$ that it can effectively approximate, thereby simplifying the prediction task for downstream blocks.
Significantly, each block produces a partial forecast $\hat{y}_l$ , which is initially aggregated at the stack level and
subsequently at the overall network level, establishing a hierarchical decomposition. The ultimate forecast
$\hat{y}$ is the summation of all partial forecasts (Oreshkin et al., 2020).
The N-BEATS model has two primary configurations: generic and interpretable. These configurations
determine how the model structures its blocks and how it processes time series data. In the generic
configuration, the model uses a stack of generic blocks that are designed to be flexible and adaptable to
various patterns in the time series data. Each generic block consists of fully connected layers with ReLU
activation functions. The key characteristic of generic configuration is its flexibility. Since the blocks are
not specialized for any specific pattern (like trend or seasonality), they can learn a wide range of patterns
directly from the data (Oreshkin et al., 2020). In the interpretable configuration, the model architecture
integrates distinct trend and seasonality components. This involves structuring the basis layers at the stack
level specifically to model these elements, allowing the stack outputs to be more easily understood.
**Trend Model:** In this stack $g_{s,l}^b$ and $g_{s,l}^f$ are polynomials of a small degree $p$, functions that vary slowly
across the forecast window, to replicate monotonic or slowly varying nature of trends:

$$\hat{y}_{s,l} = \sum_{i=0}^{p} \theta_{s,l,i}^f t^i \qquad \text{(Equation 10)}$$

The time vector $t = [0, 1, 2, \ldots , H-2, H-1]^T/H$ is specified on a discrete grid ranging from 0 to
*(H−1)/H*, projecting *H* steps into the future. Consequently, the trend forecast represented in matrix form is:

$$\hat{y}_{s,l}^{tr} = T\theta_{s,l}^f \qquad \text{(Equation 11)}$$

Where the polynomial coefficients, $\theta_{s,l}^f$, predicted by an FC network at layer *l* of stack *s*, are described by
Equations (6) and (7). The matrix *T*, consisting of powers of *t*, is represented as $[1, t, \ldots, t^p]$. When *p* is
small, such as 2 or 3, it compels $\hat{y}_{s,l}^{tr}$ to emulate a trend (Oreshkin et al., 2020).
Seasonality model: In this stack $g_{s,l}^b$ and $g_{s,l}^f$ are periodic functions, to capture the cyclical and recurring
characteristics of seasonality, such that $y_t = y_{t-\Delta}$, where $\Delta$ is the seasonality period. The Fourier series
serves as a natural foundation for modeling periodic functions:

$$\hat{y}_{s,l} = \sum_{i=0}^{\frac{H}{2}-1} \theta_{s,l,i}^f \cos(2\pi it) + \theta_{s,l,i+[H/2]}^f \sin(2\pi it) \qquad \text{(Equation 12)}$$

Consequently, the seasonality forecast is represented in the following matrix form:

$$\hat{y}_{s,l}^{seas} = S\theta_{s,l}^f \qquad \text{(Equation 13)}$$

$$S = \left[1, \cos(2\pi t), \dots, \cos\left(2\pi\left[\frac{H}{2}-1\right]t\right), \sin(2\pi t), \dots, \sin\left(2\pi\left[\frac{H}{2}-1\right]t\right)\right] \qquad \text{(Equation 14)}$$

Where the Fourier coefficients $\theta_{s,l}^f$, that predicted by an FC network at layer $l$ of stack $s$, are described by
Equations (6) and (7). The matrix $S$ represents sinusoidal waveforms. As a result, the forecast $\hat{y}_{s,l}^{seas}$
becomes a periodic function that imitates typical seasonal patterns (Oreshkin et al., 2020).

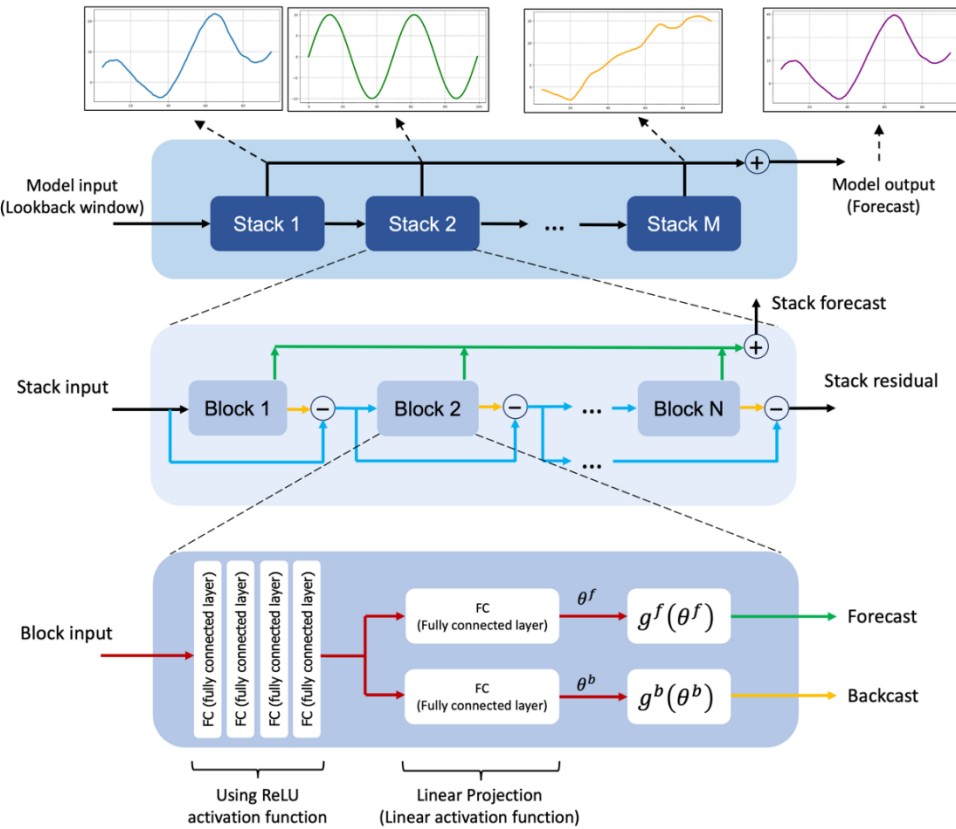


Figure 3. The N-BEATS modeling structure, used in this research.

### 2.2.3. N-HiTS

N-HiTS builds upon the N-BEATS architecture but with improved accuracy and computational efficiency for long-horizon forecasting. N-HiTS utilizes multi-rate sampling and multi-scale synthesis of forecasts, leading to a hierarchical forecast structure that lowers computational demands and improves prediction accuracy (Challu et al., 2022).

Like N-BEATS, N-HiTS employs local nonlinear mappings onto foundational functions within numerous blocks. Each block includes an MLP that generates backcast and forecast output coefficients. The backcast output refines the input data for the following blocks, and the forecast outputs are combined to generate the final prediction. Blocks are organized into stacks, with each stack dedicated to grasping specific data attributes using its own distinct set of functions. The network's input is a sequence of $L$ lags (look-back period), with $S$ stacks, each containing $B$ blocks (Challu et al., 2022).

In each block, a *MaxPool* layer with varying kernel sizes ($k_l$) is employed at the input, enabling the block to focus on specific input components of different scales. Larger kernel sizes emphasize the analysis of larger-scale, low-frequency data, aiding in improving long-term forecasting accuracy. This approach, known as multi-rate signal sampling, alters the effective input signal sampling rate for each block's MLP (Challu et al., 2022).

Additionally, multi-rate processing has several advantages. It reduces memory usage, computational demands, and the number of learnable parameters, and helps prevent overfitting, while preserving the original receptive field. The following operation is applicable to the input $y_{t-L:t,l}$ of each block, with the first block ($l = 1$) using the network-wide input, where $y_{t-L:t,1} \equiv y_{t-L:t}$.

$$y_{t-L:t,l} = MaxPool\left(y_{t-L:t,l}, k_l\right) \qquad \text{(Equation 15)}$$

In many multi-horizon forecasting models, the number of neural network predictions matches the horizon's dimensionality, denoted as $H$. For instance, in N-BEATS, the number of predictions $\left|\theta_l^f\right| = H$. This results in a significant increase in computational demands and an unnecessary surge in model complexity as the horizon $H$ becomes larger (Challu et al., 2022).

To address these challenges, N-HiTS proposes the use of temporal interpolation. This model manages the parameter counts per unit of output time ($\left|\theta_l^f\right| = \lceil r_l H \rceil$) by defining the dimensionality of the interpolation coefficients with respect to the expressiveness ratio $r_l$. To revert to the original sampling rate and predict all horizon points, this model employs temporal interpolation through the function $g$:

$$\hat{y}_{\tau,l} = g\left(\tau, \theta_l^f\right), \qquad \forall \tau \in \{t+1, \dots, t+H\}, \qquad \text{(Equation 16)}$$

$$\tilde{y}_{\tau,l} = g(\tau, \theta_l^b), \qquad \forall \tau \in \{t - L, \dots, t\}, \qquad \text{(Equation 17)}$$

$$g(\tau, \theta) = \theta[t_1] + \left( \frac{\theta[t_2] - \theta[t_1]}{t_2 - t_1} \right) (\tau - t_1) \qquad \text{(Equation 18)}$$

$$t_1 = \arg \min_{t \in \tau: t \leq \tau} \tau - t, \quad t_2 = t_1 + 1/r_l \qquad \text{(Equation 19)}$$

The hierarchical interpolation approach involves distributing expressiveness ratios over blocks, integrated
with multi-rate sampling. Blocks closer to the input employ more aggressive interpolation, generating lower
granularity signals. These blocks specialize in analyzing more aggressively subsampled signals. The final
hierarchical prediction, $\hat{y}_{t+1:t+H}$, is constructed by combining outputs from all blocks, creating
interpolations at various time-scale hierarchy levels. This approach maintains a structured hierarchy of
interpolation granularity, with each block focusing on its own input and output scales (Challu et al., 2022).
To manage a diverse set of frequency bands while maintaining control over the number of parameters,
exponentially increasing expressiveness ratios are recommended. As an alternative, each stack can be
dedicated to modeling various recognizable cycles within the time series (e.g., weekly, or daily) employing
matching $r_l$. Ultimately, the residual obtained from backcasting in the preceding hierarchy level is
subtracted from the input of the subsequent level, intensifying the next-level block's attention on signals
outside the previously addressed band (Challu et al., 2022).

$$\hat{y}_{t+1:t+H} = \sum_{l=1}^{L} \hat{y}_{t+1:t+H,l} \qquad \text{(Equation 20)}$$

$$y_{t-L:t,l+1} = y_{t-L:t,l} - \tilde{y}_{t-L:t,l} \qquad \text{(Equation 21)}$$

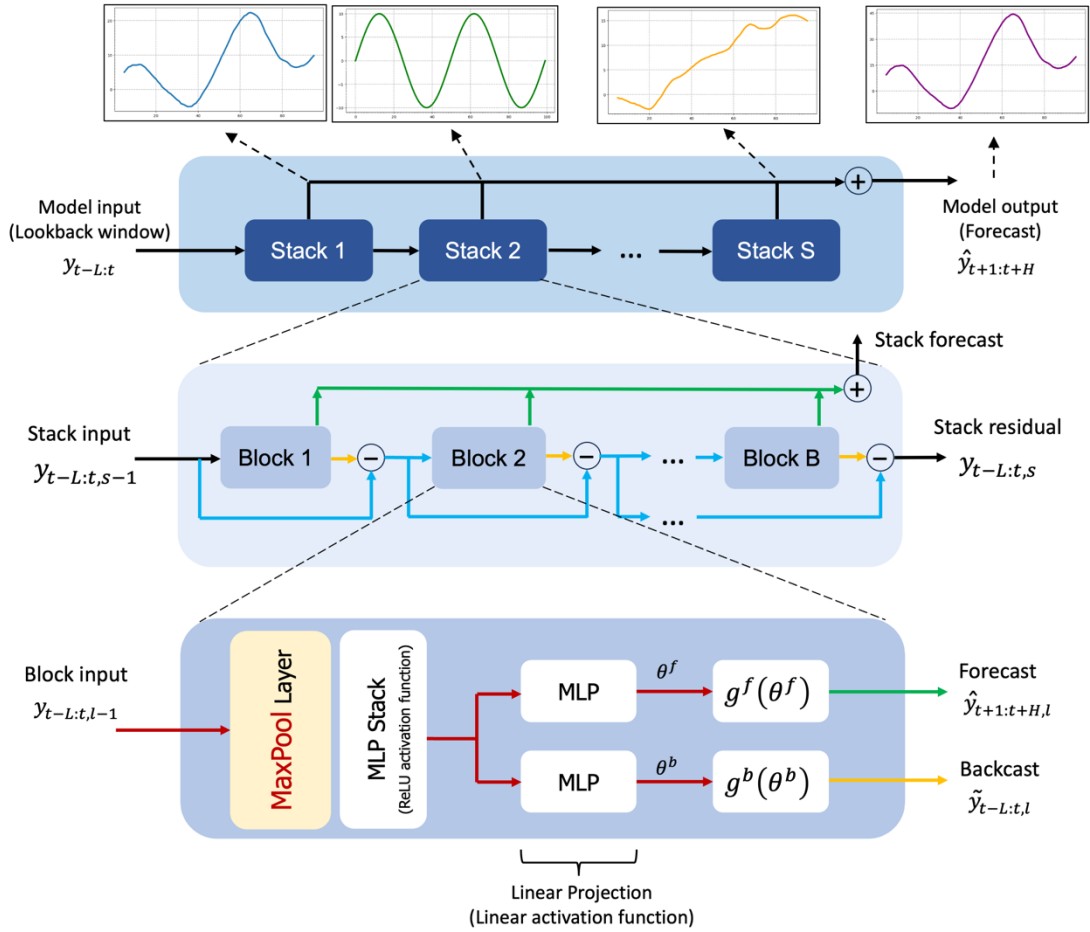

Figure 4. The structure of N-HiTS model programmed in this study. The architecture includes several Stacks, each Stack includes several Block, where each block consists of a MaxPool layer and a multi-layer which learns to produce coefficients for the backcast and forecast outputs of its basis.

## 2.3. Performance Metrics

To comprehensively evaluate the accuracy of flood predictions, we utilized a suite of metrics, including Nash-Sutcliffe Efficiency (NSE; Nash and Sutcliffe, 1970), persistent Nash-Sutcliffe Efficiency (persistent-NSE), Kling–Gupta efficiency (KGE; Gupta et al. 2009), Root Mean Square Error (RMSE), Mean Absolute Error (MAE), Peak Flow Error (PFE), and Time to Peak Error (TPE; Evin et al., 2023; Lobligeois et al., 2014). These metrics collectively facilitate a rigorous assessment of the model's performance in reproducing the magnitude of observed peak flows and the shape of the hydrograph.

NSE measures the model's ability to explain the variance in observed data and assesses the goodness-of-fit
by comparing the observed and simulated hydrographs. In hydrological studies, the NSE index is a widely
accepted measure for evaluating the fitting quality of models (McCuen et al., 2006). It is calculated as:

$$NSE = 1 - \frac{\sum_{i=1}^{n}(Q_{s_i} - Q_{o_i})^2}{\sum_{i=1}^{n}(Q_{o_i} - \overline{Q_o})^2} \qquad \text{(Equation 22)}$$

Where $Q_{o_i}$ represents observed value at time $i$, $Q_{s_i}$ represents simulated value at time $i$, $\overline{Q_o}$ is the mean
observed values and $n$ is the number of data points. An NSE value of 1 indicates a perfect match between
the observed and modeled data, while lower values represent the degree of departure from a perfect fit.
As the models are designed to predict one hour ahead in one of the prediction horizons, the persistent-NSE
is essential for evaluating their performance. The standard NSE measures the model's sum of squared errors
relative to the sum of squared errors when the mean observation is used as the forecast value. In contrast,
persistent-NSE uses the most recent observed data as the forecast value for comparison (Nevo et al., 2022).
The persistent NSE is calculated as:

$$persistent - NSE = 1 - \frac{\sum_{i=1}^{n}(Q_{s_i} - Q_{o_i})^2}{\sum_{i=1}^{n}(Q_{o_i} - Q_{o_{i-1}})^2} \qquad \text{(Equation 23)}$$

Where $Q_{o_i}$ represents the observed value at time $i$, $Q_{s_i}$ represents the simulated value at time $i$, $Q_{o_{i-1}}$ is the
observed value at the last time step $(i - 1)$ and $n$ is the number of data points.
The KGE is a widely used performance metric in hydrological modeling and combines multiple aspects of
model performance, including correlation, variability bias, and mean bias. The KGE metric is calculated
using the following equation:

$$KGE = 1 - \sqrt{(r - 1)^2 + (\alpha - 1)^2 + (\beta - 1)^2} \qquad \text{(Equation 24)}$$

Where $r$ represents Pearson correlation coefficient between observed $Q_o$ and simulated $Q_s$ values.
$\alpha$ represents bias ratio, calculated as $\alpha = \frac{\mu_s}{\mu_o}$ where $\mu_s$ and $\mu_o$ are the means of simulated and observed data,
respectively. $\beta$ represents variability ratio, calculated as $\beta = \frac{\sigma_s/\mu_s}{\sigma_o/\mu_o}$ where $\sigma_s$ and $\sigma_o$ are the standard
deviations of simulated and observed data, respectively.
RMSE quantifies the average magnitude of errors between observed and modeled values, offering insights
into the absolute goodness-of-fit, while MAE is a measure of the average absolute difference between the
modeled values and the observed values and provides a measure of the average magnitude of errors. RMSE
is calculated as:

$$RMSE = \sqrt{\frac{1}{n}\sum_{i=1}^{n}(Q_{o_i} - Q_{s_i})^2} \qquad \text{(Equation 25)}$$

and MAE is calculated as:

$$MAE = \frac{1}{n}\sum_{i=1}^{n}|Q_{o_i} - Q_{s_i}| \qquad \text{(Equation 26)}$$

Where $Q_{o_i}$ represents observed value at time $i$, $Q_{s_i}$ represents simulated value at time $i$, and $n$ is the number
of data points. RMSE and MAE provide information about the magnitude of modeling errors, with smaller
values indicating a better model fit.
PFE quantifies the magnitude disparity between observed and modeled peak flow values. The PFE metric
is defined as:

$$PFE = \frac{|Q_{o\,max} - Q_{s\,max}|}{Q_{o\,max}} \qquad \text{(Equation 27)}$$

Where $Q_{o\,max}$ represents the observed peak flow value, and $Q_{s\,max}$ signifies the simulated peak flow value.
The PFE metric, expressed as a dimensionless value, provides a quantitative measure of the relative error
in predicting peak flow magnitudes concerning the observed values. A smaller PFE denotes more accurate
modeling of peak flow magnitudes, with a value of zero indicating a perfect match.
TPE assesses the temporal alignment of peak flows in the observed and modeled hydrographs. The TPE
metric is computed as:

$$TPE = |T_{o\,max} - T_{s\,max}| \qquad \text{(Equation 28)}$$

Where $T_{o\,max}$ signifies the time at which the peak flow occurs in the observed hydrograph, and $T_{s\,max}$
represents the time at which the peak flow occurs in the simulated hydrograph. TPE that is measured in
units of time (hours), provides insight into the precision of peak flow timing. Smaller TPE values indicate
a superior alignment between the observed and modeled peak flow timing, while larger TPE values indicate
discrepancies in the temporal occurrence of peak flows.
The utilization of these five metrics, PFE, persistent-NSE, TPE, NSE, and RMSE, collectively provides a
robust and multifaceted assessment of flood prediction performance. This approach ensures that both the
magnitude and timing of peak flows, as well as the overall hydrograph shape, are accurately calibrated and
validated.

## 2.4. Sensitivity and Uncertainty Analysis

When implementing NN models, it's crucial to understand how each input feature affects the model's performance or outputs. To achieve this, we systematically excluded each input feature from the model one by one (the Leave-One-Out method). For each exclusion, we retrained the model without that specific input feature and then tested its performance against a test dataset. This method helps in understanding which input features are most critical to the model's performance and which ones have a lesser impact. It also allows us to identify any input features that may be redundant or have little effect on the overall outcome, thus potentially simplifying the model without sacrificing accuracy.

In this study, we utilized probabilistic approaches to quantify the uncertainty in flood prediction. This method is rooted in statistical techniques employed for the estimation of unknown probability distributions, with a foundation in observed data. More specifically, we leveraged the Maximum Likelihood Estimation (MLE) approach, which entails the determination of MQL objective values that optimize the likelihood function. The likelihood function quantifies the probability of MQL objective taking values, given the observed realizations.

We incorporated the MQL as a probabilistic error metric into algorithmic architecture. MQL performs an evaluation by computing the average loss for a predefined set of quantiles. This computation is grounded in the absolute disparities between predicted quantiles and their corresponding observed values. By considering multiple quantile levels, MQL provides a comprehensive assessment of the model's ability to capture the distribution of the target variable, rather than focusing solely on point estimates.

The MQL metric also aligns closely with the Continuous Ranked Probability Score (CRPS), a standard tool for evaluating predictive distributions. CRPS measures the difference between the predicted cumulative distribution function and the observed values by integrating over all possible quantiles. The computation of CRPS involves a numerical integration technique that discretizes quantiles and applies a left Riemann approximation for CRPS integral computation. This process culminates in the averaging of these computations over uniformly spaced quantiles, providing a robust evaluation of the predictive distribution $\hat{F}_t$.

$$\text{MQL}\left(Q_\tau, \left[\hat{Q}_\tau^{q_1}, \ldots, \hat{Q}_\tau^{q_i}\right]\right) = \frac{1}{n} \sum_{q_i} \text{QL}\left(Q_\tau, \hat{Q}_\tau^{q_i}\right) \qquad \text{(Equation 29)}$$

$$\text{CRPS}\left(Q_\tau, \hat{F}_\tau\right) = \int_0^1 \text{QL}\left(Q_\tau, \hat{Q}_\tau^{q_i}\right) dq \qquad \text{(Equation 30)}$$

$$\text{QL}\left(Q_\tau, \hat{Q}_\tau^q\right) = \frac{1}{H} \sum_{\tau=t+1}^{t+H} \left((1-q)\left(\hat{Q}_\tau^q - Q_\tau\right) + q\left(Q_\tau - \hat{Q}_\tau^q\right)\right) \qquad \text{(Equation 31)}$$

Where $Q_\tau$ represents observed value at time $\tau$, $\hat{Q}_\tau^q$ represents simulated value at time $\tau$, $q$ is the slope of the
quantile loss, and $H$ is the horizon of forecasting.
Implementation-wise, let $\mathcal{D} = \{(X_t, y_{t+h})\}_{t=1}^N$ denote training pairs, where $X_t$ is the past 24-h discharge
context and $y_{t+h}$ the discharge $h$ hours ahead. For a fixed horizon $h$ and quantile levels $\{\tau_k\}_{k=1}^K$, each
model $f_\theta$ outputs the vector of conditional quantiles:

$$\hat{\mathbf{Q}}_{t+h} = f_\theta(X_t) = (\hat{Q}_{t+h}^{\tau_1}, \dots, \hat{Q}_{t+h}^{\tau_K}) \in \mathbb{R}^K \qquad \text{(Equation 32)}$$

Parameters $\theta$ are learned by minimizing the multi-quantile (pinball) loss:

$$\mathcal{L}(\theta) = \frac{1}{NK} \sum_{t=1}^N \sum_{k=1}^K \rho_{\tau_k}\left(y_{t+h} - \hat{Q}_{t+h}^{\tau_k}\right), \qquad \text{(Equation 33)}$$

$$\rho_\tau(u) = \max\left(\tau u, (\tau-1)u\right) = \left(\tau - \mathbb{1}_{\{u<0\}}\right)u$$

Because $\rho_\tau$ is convex and piecewise linear, its (sub)gradient with respect to $\hat{Q}_{t+h}^\tau$ is:

$$\frac{\partial \rho_\tau(y - \hat{Q}^\tau)}{\partial \hat{Q}^\tau} = \begin{cases} -(1-\tau), & y - \hat{Q}^\tau < 0, \\ -\tau, & y - \hat{Q}^\tau > 0, \end{cases} \qquad \text{(Equation 34)}$$

enabling backpropagation (Adam) without any sampling. Thus, each quantile $\hat{Q}_{t+h}^\tau$ is a direct network
output learned to satisfy the quantile condition under $\rho_\tau$. Uncertainty intervals are formed from these
quantile predictions; for a 95% band we use $[\hat{Q}_{t+h}^{0.025}, \hat{Q}_{t+h}^{0.975}]$. The resulting bands quantify the uncertainty
conditional on $X_t$.
Incorporating MQL as a central metric in our study underscores its suitability for probabilistic forecasting,
particularly in the context of uncertainty quantification. Unlike traditional error metrics that focus on point
predictions, MQL captures both central tendencies and variability by penalizing errors symmetrically across
quantiles. This property ensures balanced and reliable assessments of the predictive distribution, ultimately
enhancing the robustness and interpretability of flood prediction models.

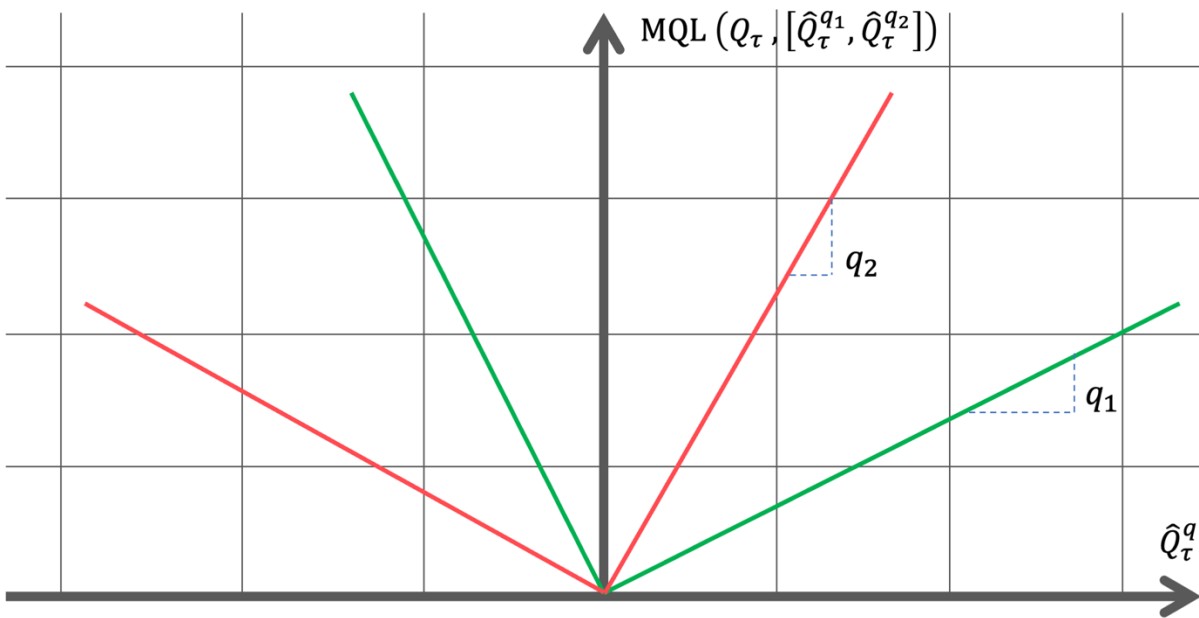


Figure 5. The MQL function which shows loss values for different values of $q$ when the true value is $Q_\tau$.
Furthermore, we employed two key indices, the R-Factor and the P-Factor, to rigorously assess the quality
of uncertainty performance in our hydrological modeling. These metrics are instrumental in quantifying the
extent to which the model's predictions encompass the observed data, thereby providing valuable insights
into the model's predictive accuracy and reliability.
The P-Factor, or percentage of data within 95PPU, is the first index used in this assessment. The P-Factor
quantifies the percentage of observed data that falls within the 95PPU, providing a measure of the model's
predictive accuracy. The P-Factor can theoretically vary from 0% to a maximum of 100%. A P-Factor of
100% signifies a perfect alignment between the model's predictions and the observed data within the
uncertainty band. In contrast, a lower P-Factor indicates a reduced ability of the model to predict data within
the specified uncertainty range.

$$P - Factor = \frac{Observations\ braketed\ by\ 95PPU}{Number\ of\ observations} \times 100 \qquad \text{(Equation 35)}$$

The R-Factor can be computed by dividing the average width of the uncertainty band by the standard
deviation of the measured variable. The R-Factor, with a minimum possible value of zero, provides a
measure of the spread of uncertainty relative to the variability of the observed data. Theoretically, the R-
Factor spans from 0 to infinity, and a value of zero implies that the model's predictions precisely match the
measured data, with the uncertainty band being very narrow in relation to the variability of the observed
data.

$$R - Factor = \frac{Average\ width\ of\ 95PPU\ band}{Standard\ deviation\ of\ measured\ variables} \times 100 \qquad \text{(Equation 36)}$$

In practice, the quality of the model is assessed by considering the 95% prediction band with the highest P-
Factor and the lowest R-Factor. This specific band encompasses most observed records, signifying the
model's ability to provide accurate and reliable predictions while effectively quantifying uncertainty. A
simulation with a P-Factor of 1 and an R-Factor of 0 signifies an ideal scenario where the model precisely
matches the measured data within the uncertainty band (Abbaspour et al., 2007).
Figure 6 shows the workflow of programming N-BEATS, N-HiTS, and LSTM for flood prediction. As
illustrated, the initial step involved cleaning and preparing the input data, which was then used to feed the
models. The workflow for each model and their output generation processes are depicted in Figure 6. We
segmented the storm events using the MIT approach, as previously described. Following this, we conducted
a sensitivity analysis using the Leave-One-Out method and performed uncertainty analysis using the MLE
approach to construct the 95PPU band. This rigorous methodology ensures a robust evaluation of model
performance under varying conditions and highlights the models' predictive reliability and resilience. We
employed the "NeuralForecast" Python package to develop the N-BEATS, N-HiTS, and LSTM models.
This package provides a diverse array of NN models with an emphasis on usability and robustness.


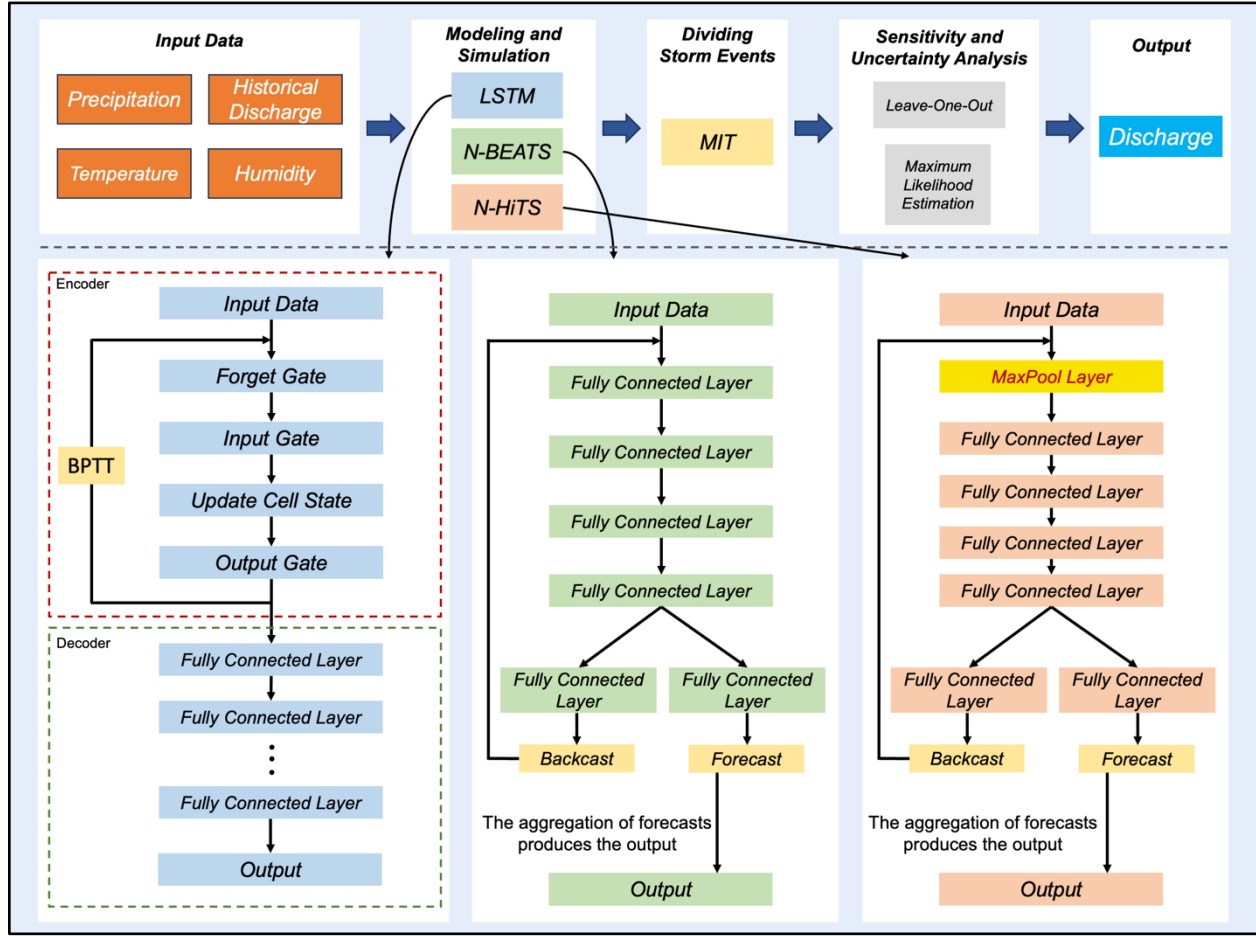


Figure 6. The workflow of N-BEATS, N-HiTS, and LSTM implementation. The upper section of the figure illustrates multiple steps from data preprocessing to model evaluation. The lower section provides a detailed view of the workflow and implementation for each model, highlighting the specific processes and methodologies employed in generating the outputs. Backpropagation Through Time (BPTT) trains LSTM by unrolling the model through time, computing gradients for each time step, and updating weights based on temporal dependencies.

## 3. Results and Discussion

### 3.1. Independent Storms Delineation

MIT's contextual delineation of storm events laid the groundwork for in-depth evaluation of rainfall events, enabling isolation and separation of rainfall events that led to significant flooding events. The nuanced outcomes of the MIT assessment contributed significantly to the understanding of rainfall variability and distribution as the dominant contributor to flood generation.

During modeling implementation, the initial imperative was the precise distinction of storm events within
the precipitation time series data of each case study. Our findings demonstrate that on average a dry period
of 7 hours serves as the optimal MIT time for both of our case studies. This outcome signifies that when a
dry interval of more than 7 hours transpires between two successive rainfall events, these subsequent
rainfalls should be considered two distinct storm events. This determination underlines the temporal
threshold necessary for distinguishing between individual meteorological phenomena in two case studies.

**3.2. Hyperparameter Optimization**
In the context of hyperparameter optimization, we systematically considered and tuned various
hyperparameters for the N-HiTS, N-BEATS, and LSTM. We searched for learning rates on a log-uniform
grid between $1 \times 10^{-4}$ and $1 \times 10^{-3}$, batch sizes {16, 32, 64}, input size {1, 6, 12, 24} hours. For the
LSTM, recurrent layers {1,2,3}, hidden units per layer {64,128,256}, activation {tanh, ReLU}, decoder
MLP depth {1,2,3}, and decoder MLP width {64,128,256} were varied during the simulation run. For N-
HiTS, stacks {2,3,4}, blocks per stack {2,3,4,5}, block MLP width {64,128,256}, and block MLP depth
{2,3,4} were explored. For N-BEATS, we searched stacks {2,3,4}, blocks per stack {2,3,4,5}, block MLP
width {64,128,256}, and block MLP depth {2,3,4}; the interpretable (trend/seasonality) basis was kept
fixed. Following extensive exploration and fine-tuning of these hyperparameters, the optimal
configurations were identified (see Table 2). For the N-HiTS model, the most favorable outcomes were
achieved with the following hyperparameter settings: 2000 epochs, "identity" for scaler type, a learning rate
of 0.001, a batch size of 32, input size of 24 hours, "identity" for stack type, 512 units for hidden layers of
each stack, step size of 1, MQLoss as loss function, and "ReLU" for the activation function. As shown in
Table 2, the N-HiTS model demonstrated superior performance with 4 stacks, containing 2 blocks each,
and corresponding coefficients of 48, 24, 12, and 1, showcasing the significance of these settings for flood
prediction.
This hyperparameter optimization was also conducted for the N-BEATS model. In this model, we
considered 2000 epochs, 3 stacks with 2 blocks, "identity" for scaler type, a learning rate of 0.001, a batch
size of 32, input size of 24 hours, "identity" for stack type, 512 units for hidden layers of each stack, step
size of 1, MQLoss as loss function, and "ReLU" for the activation function.
Moreover, the LSTM as a benchmark model yielded its best results with 5000 epochs, an input size of 24
hours, "identity" as the scaler type, a learning rate of 0.001, a batch size of 32, and "tanh" as the activation
function. Furthermore, LSTM's hidden state was most effective with two layers containing 128 units, and
the MLP decoder thrived with two layers encompassing 128 units. These meticulously optimized
hyperparameter settings represent the culmination of efforts to ensure that each model operates at its peak
potential, facilitating accurate flood prediction.
Table 2. Optimized values for the hyperparameters.

| Hyperparameter | N-HiTS | N-BEATS | LSTM |
|---|---|---|---|
| Epoch | 2000 | 2000 | 5000 |
| Scaler type | identity | identity | standard |
| Learning rate | 0.001 | 0.001 | 0.001 |
| Batch size | 32 | 32 | 32 |
| Input size | 24 hours | 24 hours | 24 hours |
| Stack type | Seasonality, trend, identity, identity | Seasonality, trend, identity | * |
| Number of units in each hidden layer | 512 | 512 | 128 |
| Loss function | MQLoss | MQLoss | MQLoss |
| Activation function | ReLU | ReLU | tanh |
| Number of stacks | 4 | 3 | * |
| Number of blocks in each stack | 2 | 2 | * |
| Stacks' coefficients | 48,24,12,1 | * | * |

*Not applicable
In Table 2, "epoch" refers to the number of training steps, and "scaler type" indicates the type of scaler used
for normalizing temporal inputs. The "learning rate" specifies the step size at each iteration while optimizing
the model, and the "batch size" represents the number of samples processed in one forward and backward
pass. The "loss function" quantifies the difference between the predicted outputs and the actual target
values, while the "activation function" determines whether a neuron should be activated. The "stacks'
coefficients" in the N-HITS model control the frequency specialization for each stack, enabling effective
handling of different frequency components in the time series data.
Another hyperparameter for all three models is input size, which is a variable that determines the maximum
sequence length for truncated backpropagation during training and the number of autoregressive inputs
(lags) that the models considered for prediction. Essentially, input size represents the length of the historical
series data used as input to the model. This variable offers flexibility in the models, allowing them to learn
from a defined window of past observations, which can range from the entire historical dataset to a subset,
tailored to the specific requirements of the prediction task. In the context of flood prediction, determining
the appropriate input size is crucial to adequately capture the meteorological data preceding the flood event.
To address this, we calculated the time of concentration ($TC$) of the watershed system and set the input size
to exceed this duration. According to the Natural Resources Conservation Service (NRCS), for typical
natural watershed conditions, the TC can be calculated from lag time, the time between peak rainfall and
peak discharge, using the formula: $Lag\ time = TC \times 0.6$ (NRCS, 2009). Specifically, the average $TC$ in
the Lower Dog River watershed and Upper Dutchmans Creek watershed was found to be 19 and 22 hours,
respectively. As these represent the average $TC$ for our case studies, we selected the 24 hours for input data,
slightly longer than the average $TC$, ensuring sufficient coverage of relevant meteorological data preceding
all flood events.

**3.3. Flood Prediction and Performance Assessment**
In this study, we conducted a comprehensive performance evaluation of N-HiTS, N-BEATS, and
benchmarked these models with LSTM, utilizing two case studies: the Lower Dog River and the Upper
Dutchmans Creek watersheds. Within these case studies, we trained and validated the models separately
for each watershed across a diverse set of storm events from 01/10/2007 to 01/10/2022 (15 years) in the
Lower Dog River and from 21/12/1994 to 01/10/2022 (27 years) in the Upper Dutchmans Creek. The
decision to train separate models for each catchment was made to account for the unique hydrological
characteristics and local features specific to each watershed. By training models individually, we aimed to
optimize performance by tailoring each model to the distinct rainfall-runoff relationship inherent in each
catchment. All algorithms were tested using unseen flooding events that occurred between 14/12/2022 and
28/03/2023. Our targets were event-focused, where operational value focuses on performance during rising
limbs, peaks, and recessions. Evaluating over the entire continuous hydrograph (testing period) can dilute
or even mask differences. For this reason, we prioritized an event-centric assessment as the primary
evaluation approach rather than full-period metrics. In the Dog River gauging station, two winter storms,
i.e., January 3rd to January 5th, 2023 (Event 1) and February 17th to February 18th, 2023 (Event 2), as well
as a spring flood event that occurred during March 26th to March 28th, 2023 (Event 3) were selected for
testing. Additionally, three winter flooding events, i.e., December 14th to December 16th, 2022 (Event 4),
January 25th and January 26th, 2023 (Event 5), and February 11th to February 13th, 2023 (Event 6), were
chosen to test the algorithms across the Killian Creek gauging station in the Upper Dutchmans Creek. The
rainfall events corresponding to these flooding events were delineated using the MIT technique discussed
in Section 3.1.
Our results for the Lower Dog River case study explicitly demonstrated the accuracy of both N-HiTS and
N-BEATS in generating the winter and spring flood hydrographs compared to the LSTM model across all
selected storm events. Although, N-HiTS prediction slightly outperformed N-BEATS during winter
prediction (January 3rd to January 5th, 2023). In this event, N-HiTS outperformed N-BEATS with a
difference of 11.6% in MAE and 20% in RMSE. The N-HiTS slight outperformance (see Tables 3 and 4)
is attributed to its unique structure that allows the model to discern and capture intricate patterns within the
data. Specifically, N-HiTS predicted flooding events hierarchically using blocks specialized in different
rainfall frequencies based on controlled signal projections, through expressiveness ratios, and interpolation
of each block. The coefficients are then used to synthesize backcast through
$\tilde{y}_t - L{:}t, l$ and forecast $(\tilde{y}_{t+1}{:}t + H, l)$ outputs of the block as a flood value. The coefficients were locally
determined along the horizon, allowing N-HiTS to reconstruct nonstationary signals over time.
While the N-HiTS emerged as the most accurate in predicting flood hydrograph among the three models,
its performance was somehow comparable with N-BEATS. The N-BEATS model exhibited good
performance in two case studies. It consistently provided competitive results, demonstrating its capacity to
effectively handle diverse storm events and deliver reliable predictions. N-BEATS has a generic and
interpretable architecture depending on the blocks it uses. Interpretable configuration sequentially projects
the signal into polynomials and harmonic basis to learn trend and seasonality components while generic
configuration substitutes the polynomial and harmonic basis for identity basis and larger network's depth.
In this study, we used interpretable architecture, as it regularizes its predictions through projections into
harmonic and trend basis that is well-suited for flood prediction tasks. Using interpretable architecture,
flood prediction was aggregated in a hierarchical fashion. This enabled the building of a very deep neural
network with interpretable flood prediction outputs.
It is essential to underscore that, despite its strong performance, the N-BEATS model did not surpass the
N-HiTS model in terms of NSE, Persistent-NSE, MAE, and RMSE for the Lower Dog River case study.
Although both models showed almost the same KGE values. Notably, the N-BEATS model showcased
superior results based on the PFE metric, signifying its exceptional capability in accurately predicting flood
peaks. However, both N-HiTS and N-BEATS models overestimated the flood peak rate of Event 2 for the
Lower Dog River watershed. This event, which occurred from February 17[th] to February 18[th], 2023, was
flashy, short, and intense proceeded by a prior small rainfall event (from February 12[th] until February 13[th])
that minimized the rate of infiltration. This flash flood event caused by excessive rainfall in a short period
of time (<8 hours) was challenging to predict for N-BEATS and N-HiTS models. In addition, predicting
the magnitude of changes in the recession curve of the third event seems to be a challenge for both models.
The specific part of the flood hydrograph after the precipitation event, where flood diminishes during a
rainless is dominated by the release of runoff from shallow aquifer systems or natural storages. It seems
both models showed a slight deficiency in capturing this portion of the hydrograph when the rainfall amount
decreases over time in the Dog River gauging station.
Conversely, in the Killian Creek gauging station, the N-BEATS model almost emerged as the top performer
in predicting the flood hydrograph based on NSE, Persistent-NSE, RMSE, and PFE performance metrics
(see Tables 3 and 4). KGE values remained almost the same for both models. In addition, both N-BEATS
and N-HiTS slightly overpredicted time to peak values for Event 5. This reflects the fact that when rainfall
varies randomly around zero, it provides less to no information for the algorithms to learn the fluctuations
and patterns in time series data. Both N-HiTS and N-BEATS provided comparable results for all events
predicted in this study. N-HiTS builds upon N-BEATS by adding a MaxPool layer at each block. Each
block consists of an MLP layer that learns how to produce coefficients for the backcast and forecast outputs.
This subsamples the time series and allows each stack to focus on either short-term or long-term effects,
depending on the pooling kernel size. Then, the partial predictions of each stack are combined using
hierarchical interpolation. This ability enhances N-HiTS capabilities to produce drastically improved,
interpretable, and computationally efficient long-horizon flood predictions.
In contrast, the performance of LSTM as a benchmark model lagged behind both N-HiTS and N-BEATS
models for all events across two case studies. Despite its extensive applications in various hydrology
domains, the LSTM model exhibited comparatively lower accuracy when tasked with predicting flood
responses during different storm events. Focusing on NSE, Persistent-NSE. KGE, MAE, RMSE, and PFE
metrics, it is noteworthy that all three models, across both case studies, consistently succeeded in capturing
peak flow rates at the appropriate timing. All models demonstrated commendable results with respect to
the TPE metric. In most scenarios, TPE revealed a value of 0, signifying that the models accurately
pinpointed the peak flow rate precisely at the expected time. In some instances, TPE reached a value of 1,
showing a deviation of one hour in predicting the peak flow time. This deviation is deemed acceptable,
particularly considering the utilization of short, intense rainfall for our analysis.
Our investigation into the performance of the three distinct forecasting models yielded compelling results
pertaining to their ability to generate 95PPU, as quantified by the P-Factor and R-Factor. These factors
serve as critical indicators for assessing the reliability and precision of the uncertainty bands produced by
the MLE. Our findings demonstrated that the N-HiTS and N-BEATS models outperformed the LSTM
model in mathematically defining uncertainty bands, in terms of R-Factor metric. The R-Factor, a crucial
metric for evaluating the average width of the uncertainty band, consistently favored the N-HiTS and N-
BEATS models over their counterparts. This finding was consistent across a diverse range of storm events.
In addition, coupling MLE with the N-HiTS and N-BEATS models demonstrated superior performance in
generating 95PPU when assessed through the P-Factor metric. The P-Factor represents another vital aspect
of uncertainty quantification, focusing on the precision of the uncertainty bands.
Figures 7 and 8 present graphical depictions of the predicted flood with 1-hour prediction horizon and
uncertainty assessment for each model as well as Flow Duration Curve (FDC) across two gauging
stations. As illustrated, the uncertainty bands skillfully bracketed most of the observational data, reflecting
the fact that MLE was successful in reducing errors in flood prediction. FDC analysis also revealed that N-
HiTS and N-BEATS models skillfully predicted the flood hydrograph, however, both models were
particularly successful in predicting moderate to high flood events (1800-6000 and >6000 cfs). In the FDC
plots, the x-axis denotes the exceedance probability, expressed as a percentage, while the y-axis signifies
flood in cubic feet per second. Notably, these plots reveal distinctive patterns in the performance of the N-
HiTS, N-BEATS, and LSTM models.
Within the lower exceedance probability range, particularly around the peak flow, the N-HiTS and N-
BEATS models demonstrated a clear superiority over the LSTM model, closely aligning with the observed
data. This observed trend is consistent when examining the corresponding hydrographs. Across all events,
the flood hydrographs generated by N-HiTS and N-BEATS exhibited a closer resemblance to the observed
data, particularly in the vicinity of the peak timing and rate, compared to the hydrographs produced by the
LSTM model. These findings underscore the enhanced predictive accuracy and reliability of the N-HiTS
and N-BEATS models, particularly in predicting moderate to high flood events as well as critical
hydrograph features such as peak flow rate and timing. The alignment of model-generated FDCs and
hydrographs with observed data in the proximity of peak flow further establishes the efficiency of N-HiTS
and N-BEATS in accurately reproducing the dynamics of flood generation mechanisms across two
headwater streams.
Table 3. The performance metrics for the Lower Dog River flood predictions with 1-hour prediction
horizon.

| Model | Performance Metric | Event 1 | Event 2 | Event 3 |
|---|---|---|---|---|
| | NSE | 0.995 | 0.991 | 0.992 |
| | Persistent-NSE | 0.947 | 0.931 | 0.948 |
| | KGE | 0.977 | 0.989 | 0.976 |
| N-HiTS | RMSE | 123.2 | 27.6 | 68.5 |
| | MAE | 64.1 | 12.0 | 37.8 |
| | PFE | 0.018 | 0.051 | 0.015 |
| | TPE (hours) | 0 | 1 | 0 |

| Model | | | | |
|---|---|---|---|---|
| | P-Factor | 96.9 % | 100 % | 93.5 % |
| | R-Factor | 0.27 | 0.40 | 0.33 |
| N-BEATS | NSE | 0.991 | 0.989 | 0.993 |
| | Persistent-NSE | 0.917 | 0.916 | 0.956 |
| | KGE | 0.984 | 0.984 | 0.98 |
| | RMSE | 154.1 | 30.5 | 62.5 |
| | MAE | 72.6 | 13.6 | 35.9 |
| | PFE | 0.0005 | 0.031 | 0.0002 |
| | TPE (hours) | 0 | 1 | 0 |
| | P-Factor | 87.8 % | 100 % | 90.3 % |
| | R-Factor | 0.17 | 0.23 | 0.24 |
| LSTM | NSE | 0.756 | 0.983 | 0.988 |
| | Persistent-NSE | -1.44 | 0.871 | 0.929 |
| | KGE | 0.765 | 0.978 | 0.971 |
| | RMSE | 841.1 | 37.9 | 79.5 |
| | MAE | 369.4 | 18.6 | 42 |
| | PFE | 0.258 | 0.036 | 0.016 |
| | TPE (hours) | 1 | 0 | 0 |
| | P-Factor | 81.8 % | 93.1 % | 96.7 % |
| | R-Factor | 0.37 | 0.51 | 0.6 |


Table 4. The performance metrics for the Killian Creek flood predictions with 1-hour prediction horizon.

| Model | Performance Metric | Event 4 | Event 5 | Event 6 |
|---|---|---|---|---|
| N-HiTS | NSE | 0.991 | 0.971 | 0.991 |
| | Persistent-NSE | 0.885 | 0.806 | 0.844 |
| | KGE | 0.982 | 0.967 | 0.991 |
| | RMSE | 28.8 | 46.0 | 19.0 |
| | MAE | 17.9 | 23.8 | 11.5 |
| | PFE | 0.017 | 0.008 | 0.020 |
| | TPE (hours) | 0 | 0 | 0 |
| | P-Factor | 92.6 % | 90.9 % | 100 % |
| | R-Factor | 0.39 | 0.48 | 0.45 |

| | | | | |
|---|---|---|---|---|
| | NSE | 0.992 | 0.973 | 0.989 |
| | Persistent-NSE | 0.908 | 0.821 | 0.823 |
| | KGE | 0.972 | 0.951 | 0.973 |
| | RMSE | 25.7 | 44.2 | 20.2 |
| N-BEATS | MAE | 18.3 | 25.9 | 14.0 |
| | PFE | 0.006 | 0.008 | 0.019 |
| | TPE (hours) | 0 | 0 | 0 |
| | P-Factor | 96.3 % | 86.3 % | 96.9 % |
| | R-Factor | 0.43 | 0.53 | 0.43 |
| | NSE | 0.952 | 0.892 | 0.935 |
| | Persistent-NSE | 0.4 | 0.27 | 0.087 |
| | KGE | 0.92 | 0.899 | 0.901 |
| | RMSE | 65.7 | 89.2 | 50.3 |
| LSTM | MAE | 41.1 | 45 | 35.9 |
| | PFE | 0.031 | 0.058 | 0.098 |
| | TPE (hours) | 1 | 0 | 0 |
| | P-Factor | 70.4 % | 72.73 % | 81.82 % |
| | R-Factor | 0.66 | 0.7 | 0.65 |


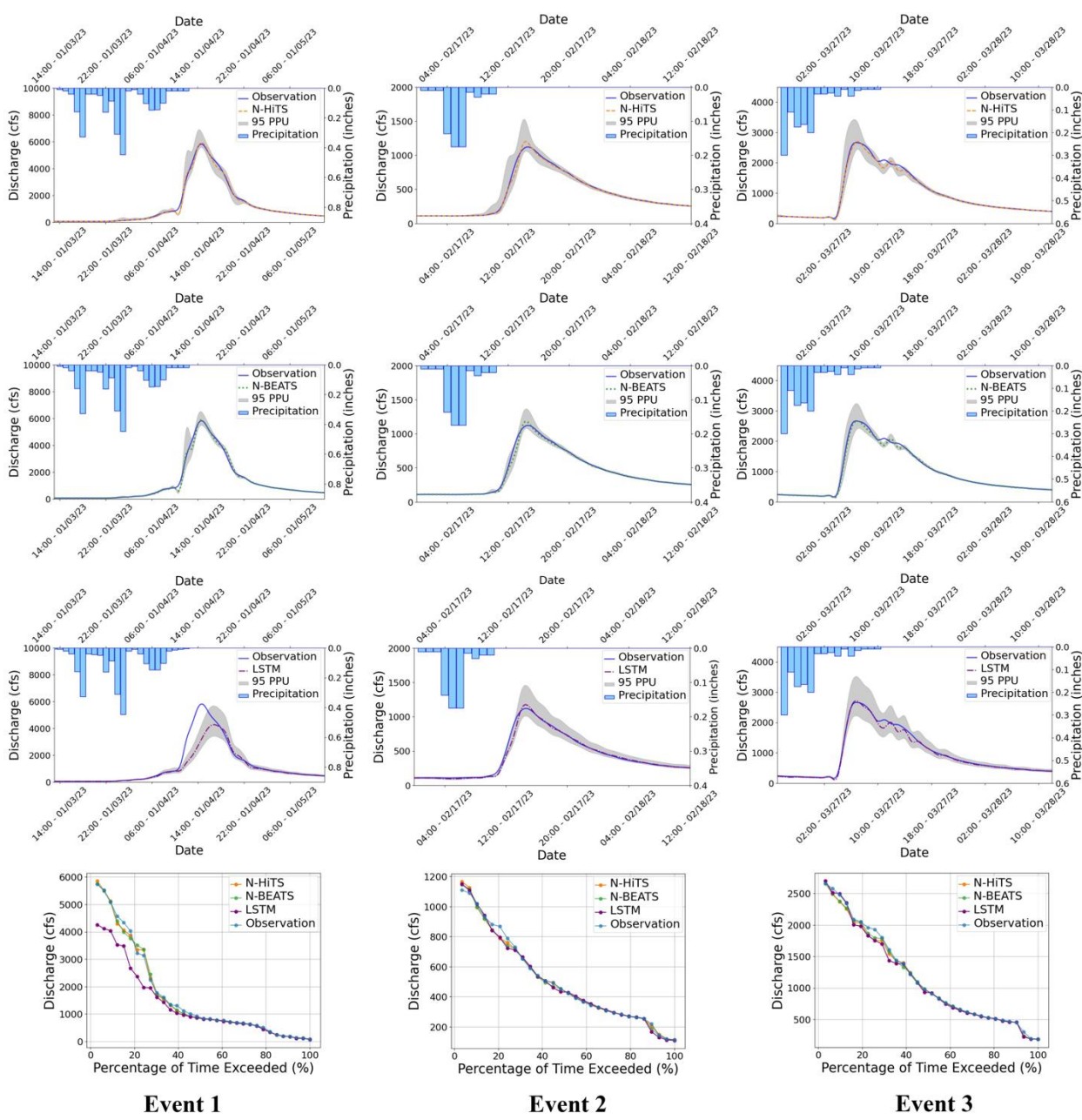

**Event 1**          **Event 2**          **Event 3**

Figure 7. 95 PPU band and FDC plots of N-HiTS, N-BEATS, and LSTM models with 1-hour prediction

horizon for the three selected flooding events in the Lower Dog River gauging station.

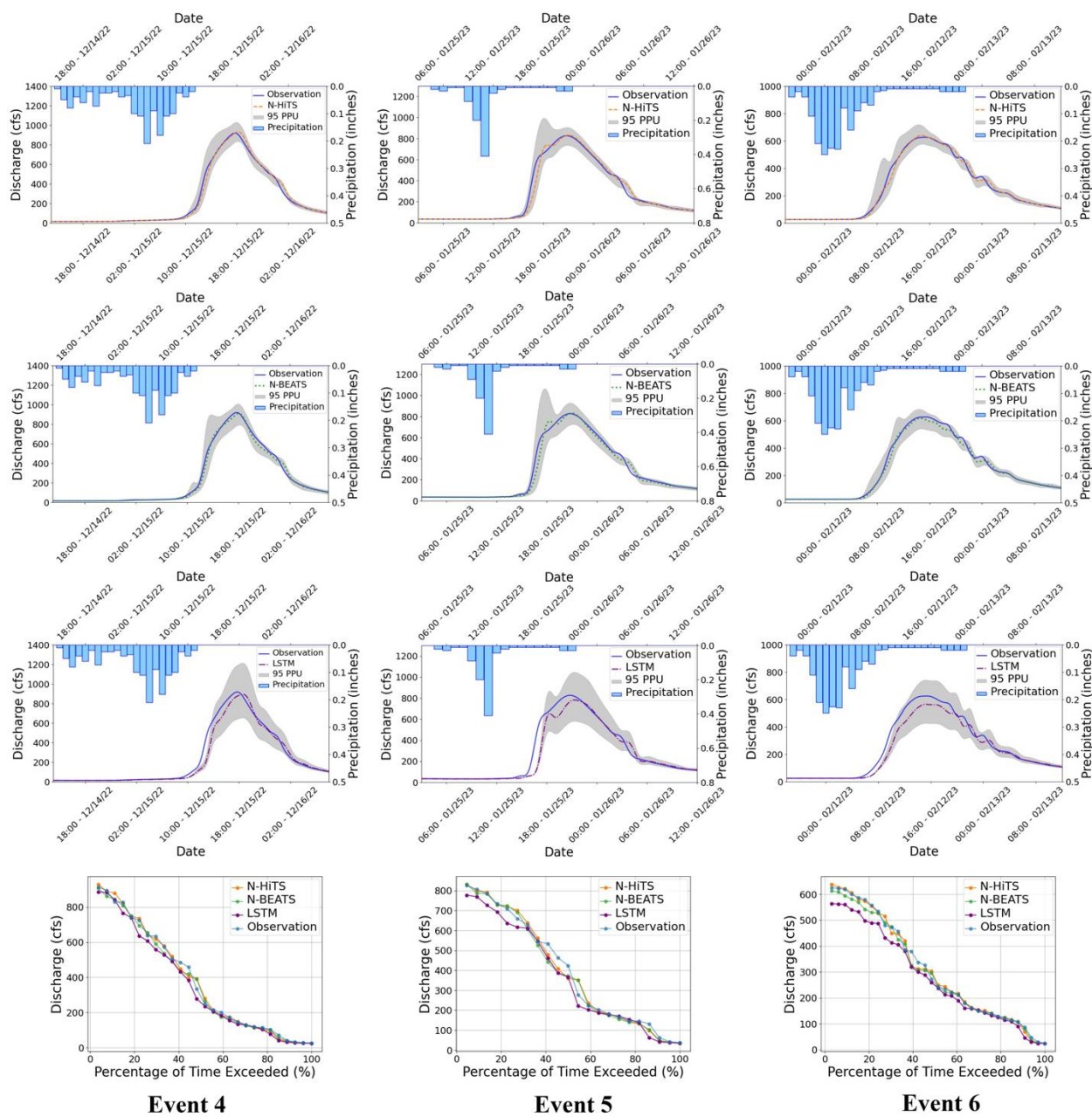

Figure 8. 95 PPU band and FDC plots of N-HiTS, N-BEATS, and LSTM models with 1-hour prediction horizon for the three selected flooding events in the Killian Creek gauging station.

To evaluate robustness across lead times, we extended the analysis to 3- and 6-hour prediction horizons. The results are presented in Figures 9-12, and Tables 5 and 6. As expected, NSE and KGE decreased while the absolute errors increased with horizon for all models; however, N-HiTS and N-BEATS continued to outperform LSTM across both stations and events. At Killian Creek station, both N-HiTS and N-BEATS preserved their lead, yielding higher NSE and lower MAE/RMSE than LSTM, while at the Lower Dog River, N-BEATS remained slightly superior on the same metrics. KGE values stayed comparable between

the two feed-forward models, and peak-focused metrics (PFE and TPE) indicated that both still captured peak magnitude and timing reliably, compared to LSTM. Uncertainty bands widened with horizon as expected, but the likelihood-based 95PPU for N-HiTS and N-BEATS maintained tighter R-Factors and competitive P-Factors relative to LSTM, especially around moderate-to-high flows. Flow-duration diagnostics at multi-hour leads reinforced these findings, showing closer alignment of N-HiTS and N-BEATS to observations in the upper tail. Overall, the multi-horizon results corroborate the 1-hour horizon results: N-HiTS and N-BEATS deliver more accurate and reliable flood forecasts than LSTM, and their relative strengths persist at 3 and 6 hours ahead. For completeness, we also evaluated 12- and 24-hour lead times. During these horizons, all models' performances declined sharply (NSE < 0.4 across sites and events), so we restrict detailed reporting to 1–6 hours where performance remains operationally meaningful.

Table 5. The performance metrics of the models with 3-hour prediction horizon.

| Model | Performance Metric | Event 1 | Event 2 | Event 3 | Event 4 | Event 5 | Event 6 |
|---|---|---|---|---|---|---|---|
| N-HiTS | NSE | 0.91 | 0.86 | 0.58 | 0.83 | 0.81 | 0.89 |
| | KGE | 0.92 | 0.92 | 0.74 | 0.85 | 0.85 | 0.88 |
| | RMSE | 506 | 107 | 485 | 122 | 119 | 65 |
| | MAE | 293 | 58 | 209 | 71 | 65 | 42 |
| | PFE | 0.03 | 0.02 | 0.08 | 0.1 | 0.07 | 0.05 |
| | TPE (hours) | 0 | 0 | 0 | 0 | 0 | 0 |
| | P-Factor | 97 % | 100 % | 93.5 % | 85 % | 72 % | 88 % |
| | R-Factor | 0.8 | 1.3 | 0.75 | 0.99 | 0.92 | 1.14 |
| N-BEATS | NSE | 0.92 | 0.88 | 0.56 | 0.82 | 0.82 | 0.89 |
| | KGE | 0.91 | 0.91 | 0.72 | 0.83 | 0.84 | 0.87 |
| | RMSE | 481 | 101 | 498 | 124 | 115 | 63 |
| | MAE | 241 | 48 | 207 | 67 | 58 | 33 |
| | PFE | 0.04 | 0.02 | 0.12 | 0.006 | 0.02 | 0.002 |
| | TPE (hours) | 1 | 0 | 2 | 0 | 0 | 0 |
| | P-Factor | 90.9 % | 93 % | 90.3 % | 92 % | 68 % | 94 % |

| | R-Factor | 0.7 | 1.2 | 0.74 | 0.78 | 1.1 | 0.87 |
|---|---|---|---|---|---|---|---|
| | NSE | 0.7 | 0.77 | 0.42 | 0.82 | 0.51 | 0.55 |
| | KGE | 0.765 | 0.87 | 0.65 | 0.79 | 0.64 | 0.69 |
| | RMSE | 928 | 139 | 575 | 125 | 190 | 133 |
| | MAE | 487 | 80 | 296 | 85 | 118 | 87 |
| LSTM | PFE | 0.12 | 0.03 | 0.16 | 0.16 | 0.44 | 0.08 |
| | TPE (hours) | 2 | 1 | 2 | 2 | 1 | 2 |
| | P-Factor | 75.8 % | 96 % | 83.9 % | 100 % | 90 % | 94 % |
| | R-Factor | 1.15 | 1.88 | 1.66 | 2.8 | 3.7 | 2.4 |


Table 6. The performance metrics of the models with 6-hour prediction horizon.

| Model | Performance Metric | Event 1 | Event 2 | Event 3 | Event 4 | Event 5 | Event 6 |
|---|---|---|---|---|---|---|---|
| | NSE | 0.82 | 0.58 | 0.51 | 0.6 | 0.7 | 0.52 |
| | KGE | 0.76 | 0.68 | 0.67 | 0.74 | 0.78 | 0.67 |
| | RMSE | 708 | 189 | 525 | 188 | 147 | 137 |
| | MAE | 423 | 90 | 257 | 110 | 90 | 77 |
| N-HiTS | PFE | 0.35 | 0.29 | 0.12 | 0.03 | 0.2 | 0.1 |
| | TPE (hours) | 2 | 3 | 0 | 0 | 3 | 3 |
| | P-Factor | 70 % | 96 % | 87 % | 92 % | 82 % | 87 % |
| | R-Factor | 0.71 | 1.1 | 1.1 | 1.8 | 1.15 | 1.2 |
| | NSE | 0.94 | 0.85 | 0.59 | 0.33 | 0.82 | 0.59 |
| | KGE | 0.83 | 0.82 | 0.73 | 0.55 | 0.79 | 0.67 |
| N-BEATS | RMSE | 386 | 112 | 481 | 244 | 115 | 126 |
| | MAE | 259 | 58 | 181 | 131 | 56 | 74 |

|  |  |  |  |  |  |  |
|---|---|---|---|---|---|---|
| **PFE** | 0.16 | 0.23 | 0.02 | 0.03 | 0.03 | 0.12 |
| **TPE (hours)** | 0 | 3 | 0 | 0 | 0 | 3 |
| **P-Factor** | 100 % | 86 % | 90.3 % | 85 % | 77 % | 78 % |
| **R-Factor** | 1.8 | 2.3 | 1.1 | 1.13 | 3.3 | 1.2 |
| **NSE** | - 0.35 | - 0.39 | - 0.22 | - 0.17 | - 0.2 | - 0.2 |
| **KGE** | 0.3 | 0.05 | 0.18 | 0.34 | 0.33 | 0.4 |
| **RMSE** | 1984 | 348 | 834 | 324 | 300 | 220 |
| **MAE** | 1304 | 192 | 468 | 234 | 201 | 174 |
| **LSTM**   **PFE** | 0.24 | 0.36 | 0.42 | 0. 6 | 0.44 | 0.42 |
| **TPE (hours)** | 3 | 4 | 3 | 0 | 2 | 2 |
| **P-Factor** | 36 % | 79 % | 90.3 % | 85 % | 86 % | 63 % |
| **R-Factor** | 1.8 | 1.9 | 2.16 | 1.6 | 3.7 | 1.6 |



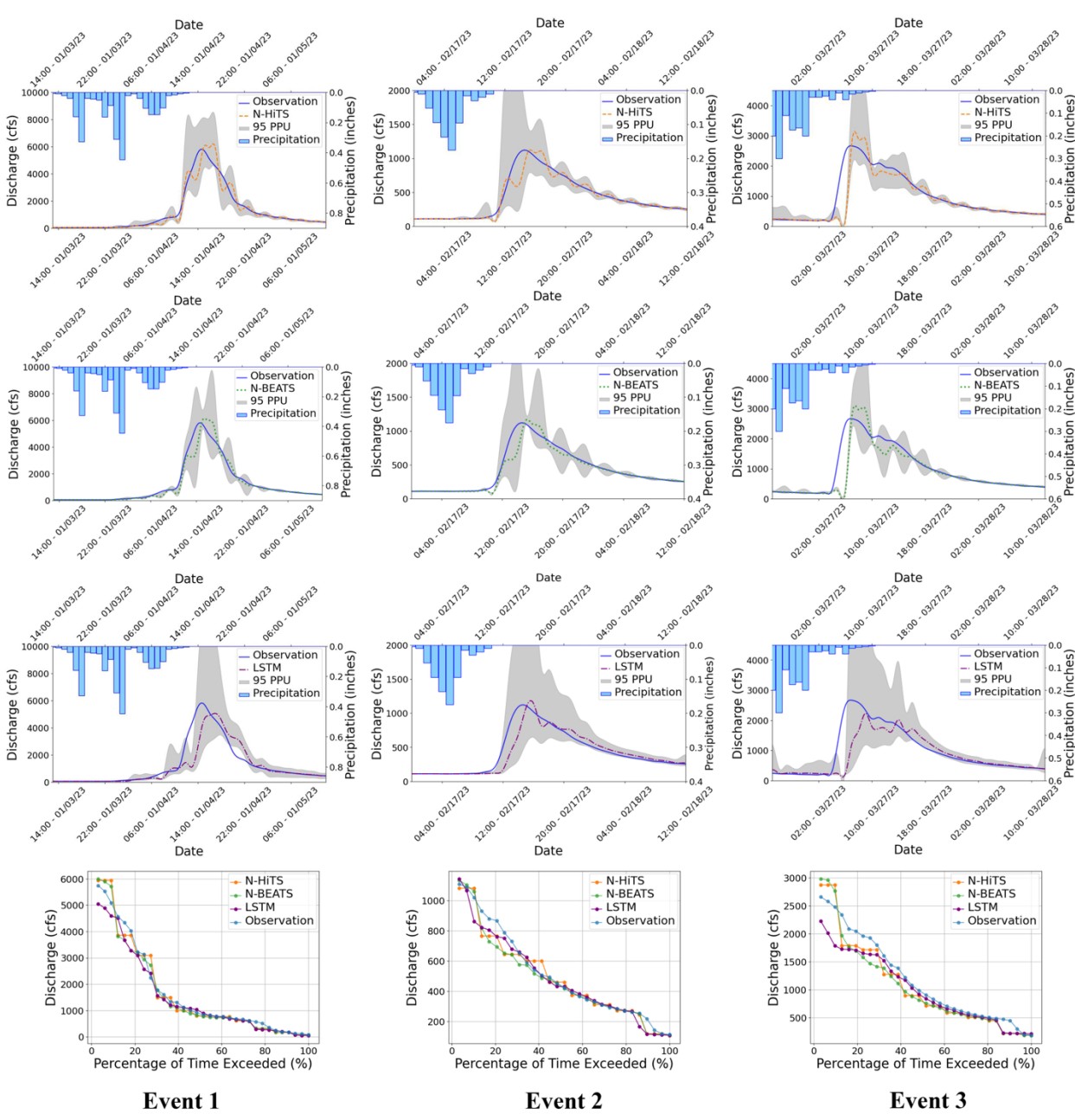


Figure 9. 95 PPU band and FDC plots of N-HiTS, N-BEATS, and LSTM models with 3-hour prediction
horizon for the three selected flooding events in the Lower Dog River gauging station.

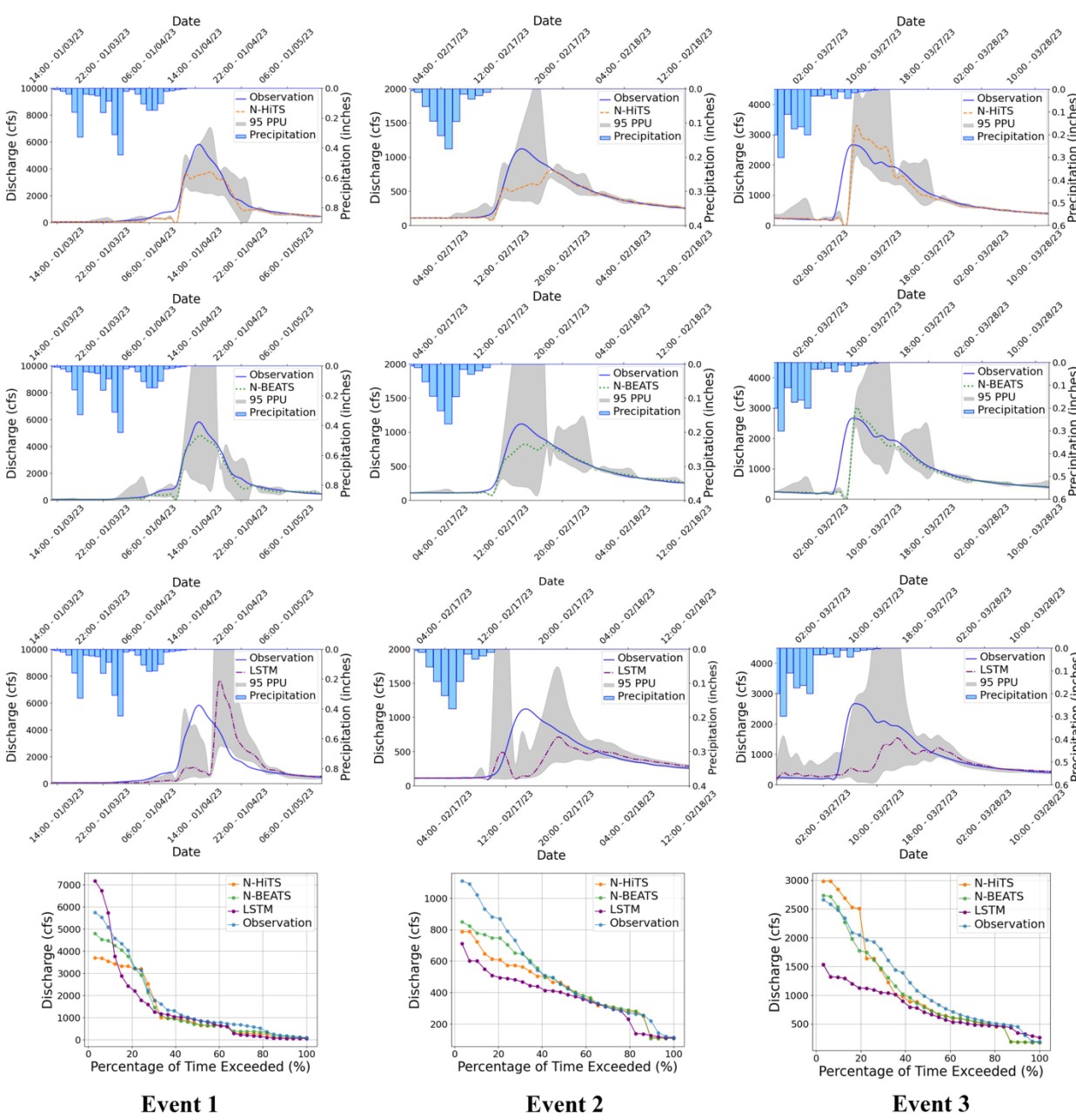

Figure 10. 95 PPU band and FDC plots of N-HiTS, N-BEATS, and LSTM models with 6-hour prediction horizon for the three selected flooding events in the Lower Dog River gauging station.

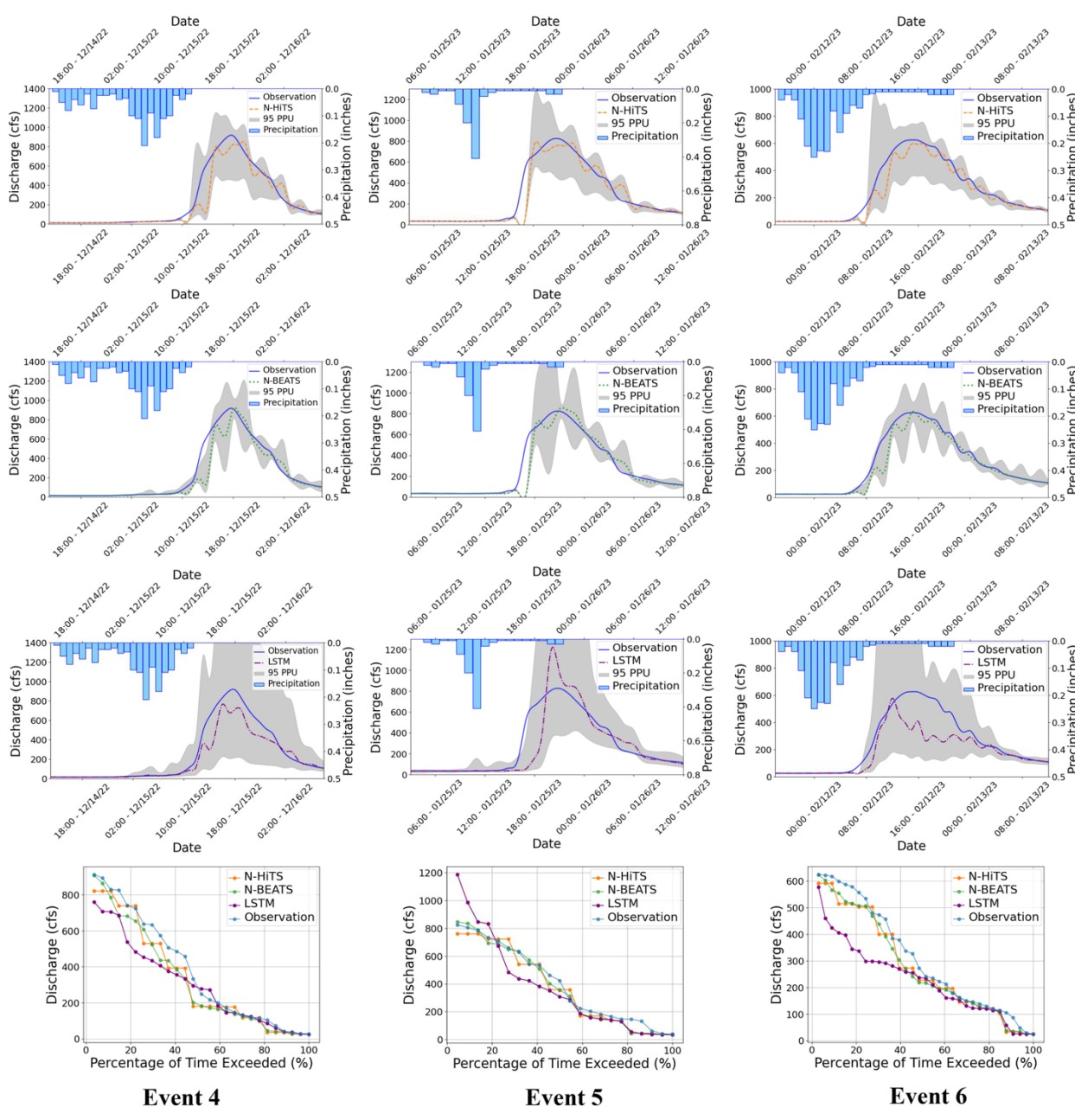

Figure 11. 95 PPU band and FDC plots of N-HiTS, N-BEATS, and LSTM models with 3-hour prediction horizon for the three selected flooding events in the Killian Creek gauging station.

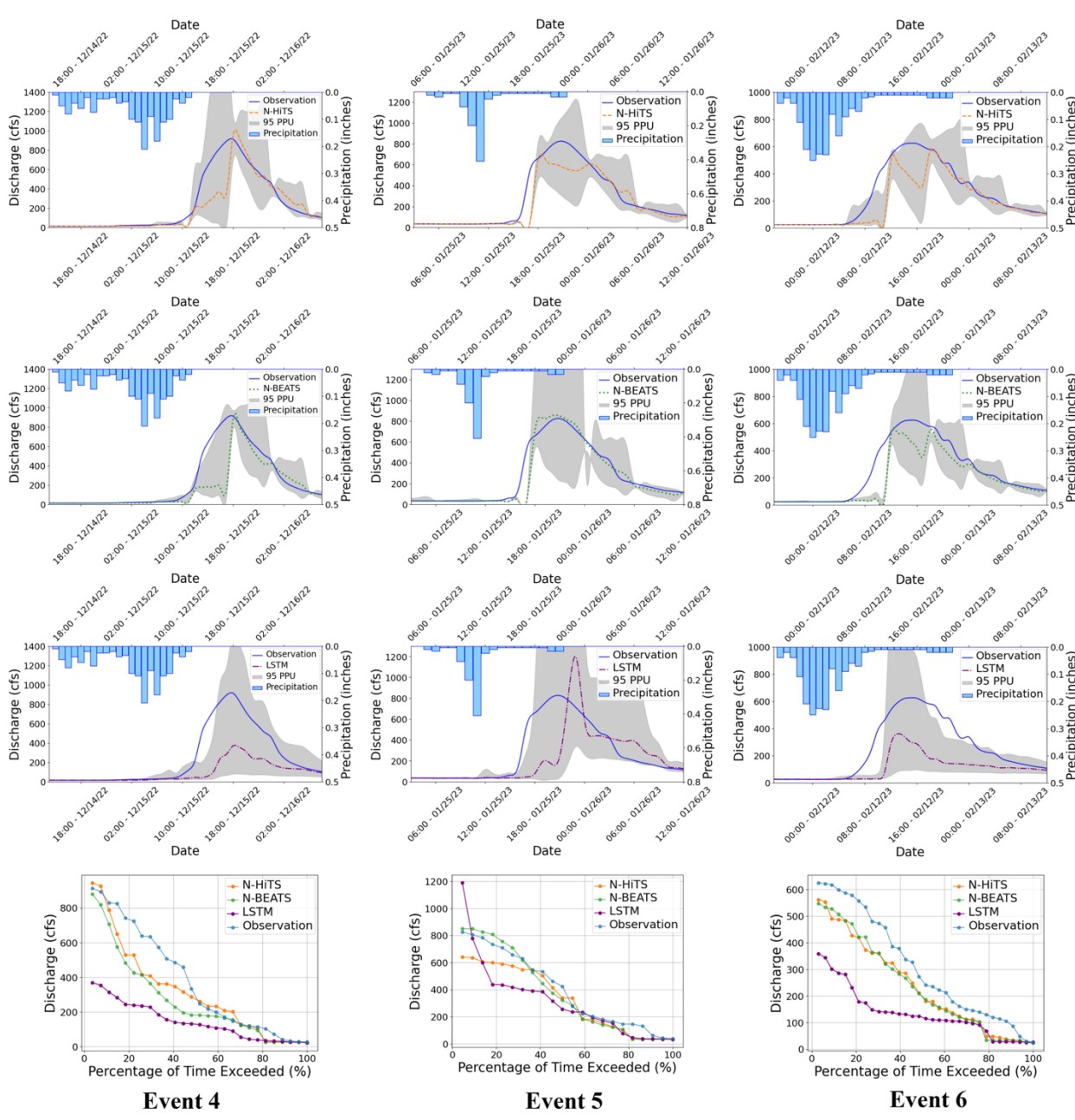


Figure 12. 95 PPU band and FDC plots of N-HiTS, N-BEATS, and LSTM models with 6-hour prediction

horizon for the three selected flooding events in the Killian Creek gauging station.

To probe cross-catchment generalizability, we trained a single "regional" model by pooling Lower Dog
River and Killian Creek, preserving per-site temporal splits and fitting a global scaler only on the pooled
training portion to avoid leakage; evaluation remained strictly per site. Relative to per-site training, pooled
fitting produced a small accuracy drop for N-HiTS and N-BEATS (~ 2 to 3 %). LSTM showed mixed
performance to pooling, it improved in some storm events but degraded in others, so that, when averaged
across both stations and storm events, LSTM's regional performance was effectively unchanged relative to
the per-site training. Despite that, the regional N-HiTS/N-BEATS matched the accuracy of the best per-site
models within the variability observed across storm events and, importantly, consistently surpassed LSTM
at both basins. Mechanistically, N-HiTS's multi-rate pooling and hierarchical interpolation, and N-
BEATS's trend/seasonality basis projection, act as catchment-invariant feature extractors that support
parameter sharing across stations.
In our investigation, we conducted an analysis to assess the impact of varying input sizes on the performance
of the N-HiTS, as the best model. We implemented four different durations as input sizes to observe the
corresponding differences in modeling performance. Notably, one of the key metrics affected by changes
in input size was 95PPU, which exhibited a general decrease with increasing input size. As detailed in Table
7, we observed a discernible trend in the R-Factor of the N-HiTS model as the input size was increased.
Specifically, there was a decline in the R-Factor as the input size expanded. This trend underscores the
influence of input size on model performance, particularly in terms of 95PPU band and accuracy.
Overall, uncertainty analysis revealed that coupling MLE with N-HiTS and N-BEATS models
demonstrated superior performance in generating 95PPU, effectively reducing errors in flood prediction.
The MLE approach was more successful in reducing 95PPU bands of N-HiTS and N-BEATS models
compared to the LSTM, as indicated by the R-Factor and P-Factor. The N-BEATS model demonstrated a
narrower uncertainty band (lower R-Factor value), while the N-HiTS model provided higher precision.
Furthermore, incorporating data with various sizes into the N-HiTS model led to a narrower 95PPU and an
improvement in the R-Factor, highlighting the significance of input size in enhancing model accuracy and
reducing uncertainty.

Table 7. N-HiTS's R-Factor results for three storm events in each case study, using 1
hour, 6 hours, 12 hours, and 24 hours input size in training.

| Input Size | 1 hour | 6 hours | 12 hours | 24 hours |
|---|---|---|---|---|
| Dog River, GA - Event 1 | 0.314 | 0.337 | 0.29 | 0.272 |
| Dog River, GA - Event 2 | 0.35 | 0.413 | 0.403 | 0.402 |
| Dog River, GA - Event 3 | 0.358 | 0.459 | 0.374 | 0.336 |
| Killian Creek, NC - Event 4 | 0.491 | 0.422 | 0.426 | 0.388 |
| Killian Creek, NC - Event 5 | 0.584 | 0.503 | 0.557 | 0.483 |
| Killian Creek, NC - Event 6 | 0.482 | 0.42 | 0.446 | 0.454 |


## 3.4. Sensitivity Analysis

In this study, we conducted a comprehensive sensitivity analysis of the N-HiTS, N-BEATS, and LSTM models to evaluate their responsiveness to meteorological variables, specifically precipitation, humidity, and temperature. The goal was to assess how the omission of input features impacts the overall modeling performance compared to their full-variable counterparts.

To execute this analysis, we systematically trained each model by excluding meteorological variables one or more at a time, subsequently evaluating their predictive performance using the entire testing dataset. The results of our analysis indicated that N-HiTS and N-BEATS models exhibited minimal sensitivity to meteorological variables, as evidenced by the negligible impact on their performance metric (i.e., NSE, Persistent-NSE, KGE, RMSE, and MAE) upon input feature exclusion.

Notably, as shown in Table 8, the performance of the N-HiTS model displayed a marginal deviation under variable omission, while the N-BEATS model exhibited consistent performance irrespective of the inclusion or exclusion of meteorological variables. The structure of this algorithm is based on backward and forward residual links for univariate time series point forecasting which does not take into account other input features in the prediction task. These findings suggest that the predictive capabilities of N-HiTS and N-BEATS models predominantly rely on historical flood data. Both models demonstrated strong performance even without incorporating precipitation, temperature, or humidity data, underscoring their ability in flood prediction in the absence of specific meteorological inputs. This capability underscores the robustness of the N-HiTS and N-BEATS models, positioning them as viable tools and perhaps appropriate for real-time flood forecasting tasks where direct meteorological data may be limited or unavailable.

Table 8. Performance metrics' values for N-HiTS, N-BEATS, and LSTM models by excluding meteorological variables one or more at a time.

| Model | Excluded Variables | NSE | Persistent-NSE | KGE | RMSE | MAE |
|---|---|---|---|---|---|---|
| **N-HiTS** | Using all variables | 0.996 | 0.92 | 0.988 | 22.66 | 4.19 |
| | Without Precipitation | 0.993 | 0.91 | 0.97 | 23.28 | 4.31 |
| | Without Humidity | 0.995 | 0.914 | 0.976 | 22.87 | 4.22 |
| | Without Temperature | 0.995 | 0.921 | 0.985 | 22.43 | 4.14 |

| | | | | | | |
|---|---|---|---|---|---|---|
| | Discharge only prediction | 0.993 | 0.911 | 0.972 | 23.21 | 4.29 |
| **N-BEATS** | Using all variables | 0.994 | 0.978 | 0.992 | 11.80 | 2.13 |
| | Without Precipitation | 0.994 | 0.978 | 0.991 | 11.86 | 2.17 |
| | Without Humidity | 0.994 | 0.978 | 0.991 | 11.81 | 2.16 |
| | Without Temperature | 0.994 | 0.978 | 0.991 | 11.82 | 2.16 |
| | Discharge only prediction | 0.994 | 0.978 | 0.991 | 11.96 | 2.17 |
| **LSTM** | Using all variables | 0.992 | 0.865 | 0.926 | 29.52 | 8.15 |
| | Without Precipitation | 0.979 | 0.665 | 0.892 | 39.46 | 19.83 |
| | Without Humidity | 0.991 | 0.843 | 0.925 | 31.73 | 9.15 |
| | Without Temperature | 0.983 | 0.628 | 0.872 | 48.95 | 11.49 |
| | Discharge only prediction | 0.976 | 0.576 | 0.692 | 52.28 | 33.5 |

**3.5 Computational Efficiency**

The computational efficiency of the N-HiTS, N-BEATS, and LSTM models, as well as a comparative analysis, is presented in Table 9. The study encompassed the entire process of training and predicting over the testing period, employing the optimized hyperparameters as previously described. Regarding the training time, it is noteworthy that the LSTM model exhibited the quickest performance. Specifically, LSTM demonstrated a training time that was 71% faster than N-HiTS and 93% faster than N-BEATS in the Lower Dog River watershed, while it was respectively,126% and 118% faster than N-HiTS and N-BEATS in the Upper Dutchmans Creek, over training dataset. This is because LSTM has simple architecture compared to the N-BEATS and N-HiTS and does not require multivariate features, hierarchical interpolation, and multi-rate data sampling. Perhaps, this outcome underscores the computational advantage of LSTM over other algorithms.

Conversely, during the testing period, the N-HiTS model emerged as the fastest and delivered the most efficient results in comparison to the other models. Notably, N-HiTS displayed a predicted time that was

33% faster than LSTM and 32% faster than N-BEATS. This finding highlights the computational efficiency
of the N-HiTS model in the context of predicting processes. Our experiments unveiled an interesting
contrast in the computational performance of these models. While LSTM excelled in terms of training time,
it lagged behind when it came to the testing period.
In the grand scheme of computational efficiency, model accuracy, and uncertainty analysis results, it
becomes evident that the superiority of the N-HiTS and N-BEATS models in terms of accuracy and
uncertainty analysis holds paramount importance. This significance is accentuated by the critical nature of
flood prediction, where precision and certainty are pivotal. Therefore, computational efficiency must be
viewed in the context of the broader objectives, with the accuracy and reliability of flood predictions taking
precedence in ensuring the safety and preparedness of the affected regions.
Table 9. Computational costs of N-HiTS, N-BEATS, and LSTM models in the Dog River and Killian
Creek gauging stations.

| Model | Training Time over Train Datasets (seconds) | | Predicting Time over Test Datasets (seconds) | |
|---|---|---|---|---|
| | Lower Dog River | Upper Dutchmans Creek | Lower Dog River | Upper Dutchmans Creek |
| N-HiTS | 256.032 | 374.569 | 1533.029 | 1205.526 |
| N-BEATS | 288.511 | 361.599 | 2028.068 | 1482.305 |
| LSTM | 149.173 | 165.827 | 2046.140 | 1792.444 |


## 4. Conclusion

This study examined multiple NN algorithms for flood prediction. We selected two headwater streams with
minimal human impacts to understand how NN approaches can capture flood magnitude and timing for
these natural systems. In conclusion, our study represents a pioneering effort in exploring and advancing
the application of NN algorithms, specifically the N-HiTS and N-BEATS models, in the field of flood
prediction. In our case studies, both N-HiTS and N-BEATS models achieved state-of-the-art results,
outperforming LSTM as a benchmark model, particularly in one-hour prediction. While a one-hour lead
time may seem brief, it is highly significant for accurate flash flood prediction particularly in an area with
a proximity to metropolitan cities, where rapid response is critical. These benchmarking results are arguably
a pivotal part of this research. However, the N-BEATS model slightly emerged as a powerful and
interpretable tool for flood prediction in most selected events.
This study focused on short-lead, operational forecasting at gauged sites, using historical discharge to
deliver robust, low-latency updates. While the evaluation is limited to two Southeastern U.S. basins, the
architecture (e.g., N-HiTS) is flexible and can incorporate additional covariates and catchment attributes.
Extending the approach to ungauged or other basins is feasible through multi-basin training and transfer
learning or few-shot adaptation when even brief warm-up records are available. These extensions represent
promising directions for future work to assess geographic transferability under the same operational
assumptions.
In addition, the results of the experiments described above demonstrated that N-HiTS multi-rate input
sampling and hierarchical interpolation along with N-BEATS interpretable configuration are effective in
learning location-specific runoff generation behaviors. Both algorithms with an MLP-based deep neural
architecture with backward and forward residual links can sequentially project the data signal into
polynomials and harmonic basis needed to predict intense storm behaviors with varied magnitudes. The
innovation in this study, besides benchmarking the LSTM model for headwater streams, was to tackle
volatility and memory complexity challenges, by locally specializing flood sequential predictions into the
data signal's frequencies with interpretability, and hierarchical interpolation and pooling. Both N-HiTS and
N-BEATS models offered similar performance as compared with the LSTM but also offered a level of
interpretability about how the model learns to differentiate aspects of complex watershed-specific behaviors
via data. The interpretability of N-HiTS and N-BEATS arises directly from their model architecture.
In the interpretable N-BEATS framework, forecasts are decomposed into trend and seasonality stacks, each
represented by explicit basis coefficients that reveal how different temporal patterns contribute to the
prediction. Similarly, N-HiTS achieves interpretability by aggregating contributions across multiple distinct
time scales, allowing insight into the temporal dynamics driving each forecast. N-HiTS aims to enhance
the accuracy of long-term time-series forecasts through hierarchical interpolation and multi-scale data
sampling, allowing it to focus on different data patterns, which prioritizes features essential to understand
flood magnitudes. N-BEATS leverages interpretable configurations with trend and seasonality projections,
enabling it to decompose time series data into intuitive components. N-BEATS interpretable architecture
is recommended for scarce data settings (such as flooding event), as it regularizes its predictions through
projections onto harmonic and trend basis.
These approaches improve model transparency by allowing understanding of how each part of the model
contributes to the final prediction, particularly when applied to complex flood patterns. Both models also
support multivariate series (and covariates) by flattening the model inputs to a 1-D series and reshaping the
outputs to a tensor of appropriate dimensions. This approach provides flexibility to handle arbitrary
numbers of features. Like LSTM, both N-HiTS and N-BEATS models support producing probabilistic
predictions by specifying a likelihood objective. In terms of sensitivity analysis, both N-HiTS and N-
BEATS maintain consistent performance even when trained without specific meteorological input.
Although, during some flashy floods, the models encountered challenges in capturing the peak flows and
the dynamics of the recession curve, which is directly related to groundwater contribution to flood
hydrograph, both models were technically insensitive to rainfall data as an input variable. This suggests the
fact that both algorithms can learn patterns in discharge data without requiring meteorological input. This
ability underscores these models' robustness in generating accurate predictions using historical flood data
alone, making them valuable tools for flood prediction, especially in data-poor watersheds or even for real-
time flood prediction when near real-time meteorological inputs are limited or unavailable. In terms of
computational efficiency, both N-HiTS and N-BEATS are trained almost at the same pace; however, N-
HiTS predicted the test data much quicker than N-BEATS. Unlike N-HiTS and N-BEATS, LSTM excelled
in reducing training time due to its simplicity and limited number of parameters.
Moving forward, it is worth mentioning that predicting the magnitude of the recession curve of flood
hydrographs was particularly challenging for all models. We argue that this is because the relation between
base flow and time is particularly hard to calibrate due to ground-water effluent that is controlled by
geological and physical conditions (vegetation, wetlands, and wet meadows) in headwater streams. In
addition, the situations of runoff occurrence are diverse and have a high measurement variance with high
frequency that can make it difficult for the algorithms to fully capture discrete representation learning on
time series.
In future studies, it will be important to develop strategies to derive analogs to the interpretable
configuration as well as multi-rate input sampling, hierarchical interpolation, and backcast residual
connections that allow for the dynamic representation of flood times series data with different frequencies
and nonlinearity. A dynamic representation of flood time series is, at least in principle, possible by
generating additive predictions in different bands of the time-series signals, reducing memory footprint and
compute time, and improving architecture parsimony and accuracy. This would allow the model to "learn"
interpretability and hierarchical representations from raw data to reduce complexity as the information
flows through the network.
While a single station provides valuable localized information, particularly for small, headwater streams
where runoff closely follows immediate meteorological conditions, it may not capture the spatial
heterogeneity of larger watersheds. In our study, the applied methods successfully captured runoff
magnitude and dynamics in small basins for an operational setting. However, broader spatial coverage and
distributed data would likely enhance model accuracy for larger regions. Consequently, our conclusions are
specifically scoped to the selected basins and forecast horizons, and broader generalizations would require
multi-region investigations in future work.
Finally, the performance of N-HiTS, N-BEATS, or other neural network architectures could be further
enhanced with robust uncertainty quantification. Approaches such as Bayesian Model Averaging (BMA)
with fixed or flexible priors (Samadi et al., 2020) or Markov Chain Monte Carlo (MCMC) optimization
methods (Duane et al., 1987) could capture both aleatoric and epistemic uncertainties. We leave these
strategies for future exploration in the context of neural flood time-series prediction.

## 5. Acknowledgements

This research is supported by the US National Science Foundation Directorate of Engineering (Grant #
CMMI 2125283; CBET 2429082). All opinions, findings, and conclusions or recommendations expressed
in this material are those of the authors and do not necessarily reflect the views of the NSF. The authors
acknowledge and appreciate Thorsten Wagener (University of Potsdam, Germany) discussion and feedback
on this manuscript. Clemson University (USA) is acknowledged for generous allotment of computing time
on the Palmetto cluster.

## 6. Open Research

The historical discharge data used in this study are from the USGS
(https://waterdata.usgs.gov/nwis/uv/?referred_module=sw), meteorological data from USDA
(https://www.ncdc.noaa.gov/cdo-web/datatools/lcd). We have uploaded the datasets and codes
used in this research to Zenodo, accessible via https://zenodo.org/records/13343364. For
modeling, we used the NeuralForecast package (Olivares et al., 2022), available at:
https://github.com/Nixtla/neuralforecast.

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
