# Peer review of "Probabilistic Hierarchical Interpolation and Interpretable Neural Network Configurations for"

_Hydrology and Earth System Sciences, 2024_

## Author Comment (AC1)

We thank the review for the positive and constructive feedback. Please find below the answers to your questions. All answers will be implemented in the revised version of the manuscript. Our reply to each of your questions/suggestions can be found below.

**Answers to Anonymous Referee #1's comments:**

**Comment 1:** *Are precipitation, temperature, and humidity enough as input variables for your neural networks?*

**Authors' answer:** In this study, precipitation, temperature, and humidity are the only variables available to develop the models, hence they were selected as primary input variables based on their direct impact on flood generation processes. The sensitivity analysis that we have performed and presented in Section 3.4 of the discussion manuscript, showed that N-HiTS and N-BEATS models maintained high performance even when individual meteorological variables were excluded, indicating the models' robustness to input variations. This robustness likely stems from the models' ability to learn essential patterns from historical discharge data alone, particularly the rainfall-runoff relationships in headwater streams. We will add more clarity in that section to reflect that this is the main conclusion and this is the reason why the selected variables are enough.

**Comment 2:** *The forcing station is a single point in the watershed while the runoff generation should be attributed to the water convergence involving a large area of the watershed, do you think a single station can represent these complex processes at large areas?*

**Authors' answer:** Thank you for this important remark. Indeed, a single station may not fully represent the spatial heterogeneity of larger watersheds. We acknowledge that using a single station can provide localized information and data for small watershed, such as headwater streams where runoff generation is more responsive to immediate meteorological conditions. However, for the particular case we have applied the methods to, capture significant runoff and flood dynamics in these streams, with a single station. Therefore, we will mention these important aspects in the conclusion section of the paper, especially that for broader areas, more spatially distributed data could improve model accuracy.

**Comment 3:** *You mentioned, your models predicted one hour ahead? Is this meaningful for flood prediction? In other words, is this enough time to escape once people know the flood will arrive one hour later.*

**Authors' answer:** You raised a valid point. Although a one-hour lead time may seem brief, it is meaningful in the context of flash flood prediction for headwater streams, where rapid response is essential. Although, we tested the model to forecast up to 72 hours in advance based on National Weather Service near real time river forecast data. We found that one-hour lead time provides the

best near real time performance. We will include this remark in the conclusion part of the revised manuscript, as this is an important remark for the readers, thank you for your comment.

**Comment 4:** *Did you train each NN model for each watershed? Trained based on one watershed and then transferred to the other one? Or trained both watersheds together?*

**Authors' answer:** Thank you for your comment. Each neural network model was trained separately for each watershed to account for local hydrological characteristics. The independent training approach enabled the models to capture watershed-specific runoff generation and flood dynamics, optimizing prediction accuracy within each unique environment. Transfer learning was not implemented. We will include this remark in the new version of the manuscript to make it clear for the readers.

---

## Author Response (AR1)

We thank the review for the positive and constructive feedback. Please find below the answers to your questions. All answers will be implemented in the revised version of the manuscript. Our reply to each of your questions/suggestions can be found in blue below.

Remark: All references to lines in the revised manuscript refer to the version with track-changes,

**Answers to Anonymous Referee #1's comments:**

**Comment 1:** *Are precipitation, temperature, and humidity enough as input variables for your neural networks?*

**Authors' answer:** In this study, precipitation, temperature, and humidity are the only variables available to develop the models, hence they were selected as primary input variables based on their direct impact on flood generation processes. The sensitivity analysis that we have performed and presented in Section 3.4 of the discussion manuscript, showed that N-HiTS and N-BEATS models maintained high performance even when individual meteorological variables were excluded, indicating the models' robustness to input variations. This robustness likely stems from the models' ability to learn essential patterns from historical discharge data alone, particularly the rainfall-runoff relationships in headwater streams. We have added additional performance metrics to Section 3.4 of the updated manuscript, and clarified that both models exhibited strong performance, even without the inclusion of precipitation, temperature, or humidity data (lines 771-772, of the manuscript with track-changes version).

**Comment 2:** *The forcing station is a single point in the watershed while the runoff generation should be attributed to the water convergence involving a large area of the watershed, do you think a single station can represent these complex processes at large areas?*

**Authors' answer:** Thank you for this important remark. Indeed, a single station may not fully represent the spatial heterogeneity of larger watersheds. We acknowledge that using a single station can provide localized information and data for small watershed, such as headwater streams where runoff generation is more responsive to immediate meteorological conditions. However, for the particular case we have applied the methods to, capture significant runoff and flood dynamics in these streams, with a single station. The new version of the manuscript incorporated these important aspects, especially that for broader areas, more spatially distributed data could improve model accuracy. Moreover, additional discussion has been incorporated into the conclusion section, specifically in lines 973–978 of the reviewed version of the manuscript, the track-changes version of it.

**Comment 3:** *You mentioned, your models predicted one hour ahead? Is this meaningful for flood prediction? In other words, is this enough time to escape once people know the flood will arrive one hour later.*

**Authors' answer:** You raised a valid point. Although a one-hour lead time may seem brief, it is meaningful in the context of accurate flash flood prediction particularly in an area with a proximity to large metropolitan cities, where rapid response is critical. Although, we tested the model to forecast up to 72 hours in advance based on National Weather Service near real time river forecast data. We found that one-hour lead time provides the best near real time performance. We included this in the conclusion part of the revised manuscript, as this is an important remark for the readers. Additional discussion has been incorporated into the conclusion section, specifically in lines 905–907, of the reviewed version of the manuscript.

**Comment 4:** *Did you train each NN model for each watershed? Trained based on one watershed and then transferred to the other one? Or trained both watersheds together?*

**Authors' answer:** Thank you for your comment. Each neural network model was trained separately for each watershed to account for local hydrological characteristics. The independent training approach enabled the models to capture watershed-specific runoff generation and flood dynamics, optimizing prediction accuracy within each unique environment. Transfer learning was not implemented. We included this remark in the new version of the manuscript in the result section (lines 590–594, of the revised manuscript, please see track-changes version).

We thank the review for the constructive feedback. Please find below the answers to your questions. All answers are implemented in the revised version of the manuscript. Our reply to each of your questions/suggestions can be found in blue below.

Remark: All references to lines in the revised manuscript refer to the version with track-changes,

**Answers to Anonymous Referee #2's comments:**

**Comment 1:** *Interpretability and Model Complexity: The paper claims that N-HiTS and N-BEATS models offer interpretability. However, further elaboration on how these models achieve interpretability would strengthen the paper. Including visual examples or providing a more explicit breakdown of how interpretability manifests in model outputs could clarify this for readers who may be less familiar with these architectures.*

**Authors' answer:** Thank you for pointing out that interpretability and model complexity are not sufficiently explained in the manuscript. In the new version of the manuscript, we enhanced this portion. The discussion has been added throughout the paper, specifically in lines 920–931. Additionally, the architecture figures for the N-HiTS and N-BEATS models have been modified/updated to better represent the interpretability of these models in lines 320 and 364.

**Comment 2:** *Hyperparameter Selection: The selection process for critical hyperparameters like the lookback window size is not fully justified. Lookback windows are crucial in sequence-based forecasting, and this choice should either be explored as a hyperparameter or explained in greater detail, particularly given the model's dependency on residuals for subsequent window predictions. Additionally, since a 24-hour lookback window is used, further elaboration on how this length captures relevant hydrological features, like seasonality or trends, would enhance clarity.*

**Authors' answer:** We agree with your assessment, and we clarified this in the revision. The selection of a 24-hour lookback window was guided by the average Time of Concentration (TC) values for the particular watersheds under study, where the average TC were close to 19 hours in the Lower Dog River watershed, and 22 hours in the Upper Dutchmans Creek watershed. By setting the lookback window to 24 hours, the model could capture essential meteorological data preceding flood events, reflecting both short-term variances and any potential longer trends relevant to hydrological processes. We also evaluated different lookback window sizes (input sizes) from 1 hour to 24 hours to analyze the impact of this hyperparameter on the results. All these mentioned important aspects are included in the results section of the paper, specifically in lines 582–584.

**Comment 3:** *Metrics Selection: While NSE, RMSE, and MAE are utilized, the omission of the Kling-Gupta Efficiency (KGE) index is notable. KGE is especially relevant for flood forecasting*

*as it provides insights into peak flow timing, magnitude, and correlation. Including KGE would add robustness to the evaluation by capturing aspects critical to hydrological modeling.*

**Authors' answer:** Thank you for this suggestion, which we find it very valuable. The inclusion of Kling-Gupta Efficiency (KGE) can certainly strengthen the model evaluations. The KGE metric has been added as one of the performance metrics in the methodology section in lines 390-396. All tables containing performance metrics in the results section have been updated accordingly.

**Comment 4:** *Interpretability in Model Outputs: Although the paper claims interpretability for both N-HiTS and N-BEATS, the explanation is somewhat abstract. Providing visual aids or case studies that illustrate interpretability in flood prediction contexts would be beneficial. Specifically, the paper mentions that projections onto harmonic and trend bases improve prediction accuracy, but further clarification on the physical interpretability of these projections would help. Given the use of a 24-hour window, it would be helpful to explain whether trends, network depth, or some other feature captures seasonality and why this choice is appropriate for flood prediction.*

**Authors' answer:** You raised a valid point. Both N-HiTS and N-BEATS capture trends and seasonality through basis functions in the interpretable configuration. For flood prediction, these components allow models to project periodic and steady trends, enhancing physical interpretability. Additional discussion has been incorporated into the conclusion section, specifically in lines 920–931.

**Comment 5:** *Uncertainty Analysis: The application of Maximum Likelihood Estimation (MLE) for uncertainty quantification is intriguing. However, more details on how MLE is applied in this context would improve reproducibility. A clearer formulation of MLE within the training process or its integration with multi-quantile loss could better inform readers about the strengths and limitations of this approach. Additionally, bootstrapping methods could help quantify uncertainty and assess whether observed performance differences between models are statistically significant, providing a more robust comparison.*

**Authors' answer:** The MLE was implemented by optimizing the likelihood function to capture prediction distribution characteristics. We acknowledge that a clearer presentation of how MLE integrates with the multi-quantile loss in training can enhance reproducibility, while bootstrapping could provide additional quantification of model prediction variability. Given the paper scope and length incorporating multiple uncertainty quantification methods would lose the focus of the presented content. The authors are already conducting a separate study on different uncertainty quantification methods for these models, with the aim to publish the findings in a subsequent paper. Additional clarification about MQL has been added into the methodology section, specifically in lines 440-465. We appreciate the comment.

**Comment 6:** *Separate Model Training for Each Catchment: Each model was trained separately for each catchment, rather than training a single model on both catchments. This approach limits the assessment of the models' generalizability across different hydrological conditions. Training a unified model on data from both catchments would provide insights into the model's adaptability and robustness across diverse environments, which is crucial for broader flood prediction applications. I recommend including an analysis of a single model trained across both catchments to evaluate cross-catchment performance.*

**Authors' answer:** Thank you for the comment. We included in the revised manuscript the below explanation:

"The decision to train separate models for each catchment was made to account for the unique hydrological characteristics and local features specific to each watershed. By training models individually, we aimed to optimize performance by tailoring each model to the distinct rainfall-runoff relationship inherent in each catchment." Additional clarification has been added into the result section, specifically in lines 590–594.

**Comment 7:** *Data Splits for Training, Validation, and Testing: It appears the observational data up to October 1, 2022, was used for training, and data from October 1, 2022, to March 28, 2023, was used for validation. However, the absence of an unseen test set to demonstrate generalization capabilities raises concerns. Dividing the dataset into three splits (training, validation, and testing) would allow for hyperparameter optimization on the validation set and final results on an unseen test set, demonstrating the model's generalization. Including metrics like loss curves for the training and validation sets or evaluation metrics on a test set would help assess model performance and detect overfitting thereby enhancing reliability.*

**Authors' answer:** Thank you for this direction. In our study, the models were trained and validated on data up to October 1, 2022. We then used data from October 1, 2022, onward as an unseen test set to evaluate the models' forecasting capabilities. This approach allowed us to optimize hyperparameters during training and validation while ensuring the final performance metrics reflected the models' ability to predict on unseen data. The clarification has been added into the result section, specifically in lines 588–590.

**Comment 8:** *Model Reproducibility: Simplifying the explanation of the Multi-Quantile Loss (MQL) function could make the methodology more accessible. Additionally, code availability or pseudocode in an appendix would enhance reproducibility and facilitate further exploration by other researchers.*

**Authors' answer:** We agree that simplifying the explanation of the Multi-Quantile Loss (MQL) function would improve accessibility for readers. The clarification about MQL has been added into the methodology section in lines 440-465. For the open research section, we mentioned that the source code for this study will be available after publication of the paper results in a repository, specifically Zenodo.

**Comment 9:** *Input Sensitivity Inconsistency (Line 568-569): The statement here suggests that the models are indeed sensitive to input conditions, especially during extreme events. However, in the following section, the paper concludes that the models are not sensitive to input data, which presents an inconsistency. This contradiction should be addressed.*

**Authors' answer:** Thanks for your comments. In these sections of the manuscript, the model results indicate challenges in capturing peak rates during flashy floods, which represent anomalies in discharge and deviate from typical rainfall-response patterns in the time series data. In addition, Lines 568-569 discussed the deficiency of both N-BEATS and N-HiTS models in capturing the dynamics of the recession curve which is directly related to groundwater contribution to flood hydrograph. However, both models are technically insensitive to rainfall data as an input variable, suggesting they can learn from discharge patterns (which inherently include precipitation effects) without requiring meteorological data. Since the models are trained on regular discharge patterns, they encounter difficulties to capture the peak rates and the recession curve due to short duration, intense runoff as well as a shallow aquifer/groundwater contribution. This discussion and clarification have been incorporated into the conclusion section, specifically in lines 936–941.

---

## Author Response (AR2)

We sincerely thank the reviewer for the thoughtful and constructive feedback. Below, we provide detailed responses to each comment, along with explanations of how and where the corresponding revisions have been incorporated into the manuscript with track changes. Line numbers may vary slightly depending on formatting.

**Anonymous Referee #3's comments:**

*The authors applied two recently proposed neural networks N-HiTS and N-BEATS to perform rainfall-runoff modeling. They demonstrated the applications of the two models at two watersheds and their improved performance over the widely adopted LSTM model. Leave-one-out was employed to perform sensitivity analysis on the model. A probabilistic loss function was used to account for the prediction uncertainty. Despite the success, the current setup is insufficient to conclude that "both N-HiTS and N-BEATS demonstrated significant performance improvements over the LSTM benchmark". Therefore, I suggest a major revision with the following main comments and minor suggestions.*

**Authors' answer:** Thank you for this summary and for highlighting the claims of model performance. To make the manuscript more comprehensive and accessible, we have added a discussion of a single pooled regional model as well as forecasts at longer horizons (3- and 6-hour). While the original manuscript focused on the operational 1-hour use case to maintain a clear narrative, the pooled two-site model and extended-horizon forecasts were initially treated as ancillary analyses. In response to the reviewer's comment, we have now incorporated these results into the main text, presenting them clearly and unambiguously in Lines 707–717.

**Main comments:**
**1:** *Two watersheds in the southeast US are not representative enough to demonstrate the better performance of N-HiTS and N-BEATS. In Kratzert et al. (2018) and related studies, LSTM has been applied to several hundred catchments, e.g., using CAMELS dataset, and compared to traditional hydrological models. Thus, the authors are recommended to apply both models at watersheds from other locations (e.g., using CAMELS or additional USGS gages across the country) to generalize their conclusion.*

**Authors' answer:** We thank the reviewer for this important suggestion. The primary objective of our study was to evaluate model capabilities under an operational, hourly forecasting setting, rather than conducting a nationwide benchmarking. Consequently, our design differs from CAMELS-based studies (e.g., Kratzert et al., 2018), which predominantly focus on daily-timestep modeling.

We selected two hydrologically contrasting southeastern U.S. basins with high-quality, continuous records to enable a controlled comparison. This approach minimizes confounding factors and allows us to test whether the architectural features of N-HiTS and N-BEATS—such as multi-scale

structures and component-wise decomposition—yield performance gains across distinct runoff regimes within a shared climatic region.

We fully agree that results from only two basins are insufficient to claim CONUS-wide superiority. Accordingly, we have revised our manuscript to frame the conclusion as evidence that N-HiTS and N-BEATS outperform a well-tuned LSTM in these two contrasting watersheds under an hourly operational setting. Broad generalization to other regions is beyond the scope of this study. As future work, we plan to extend the analysis to a multi-region, multi-basin dataset (e.g., additional USGS gages at hourly resolution) to evaluate geographic transferability while maintaining the same operational assumptions. This clarification has been added to the revised manuscript (Lines 798-804).

**2:** *The 3-month test period is too short to show that N-HiTS and N-BEATS were better than LSTM, while 10~20 years of data were used in training. I would recommend that (1) using at least two years for test period that are separate from the training; and (2) demonstrating the NSE/KGE/MAE/etc performance over the entire test period, in addition to the flood events.*

**Authors' answer:** Thank you for the suggestion and the opportunity to clarify our evaluation design. Models were trained and validated on data up to October 1, 2022, and evaluated on an unseen hourly test window spanning October 1, 2022, to April 1, 2023. Within this period, our analysis focused on flood events, highlighting three representative events for direct side-by-side comparison.

Our study emphasizes event-focused, hourly flood forecasting, where operational value is concentrated on the rising limbs, peaks, and recessions. Evaluating performance over the entire continuous hydrograph—dominated by non-flood periods—can dilute or mask differences precisely at the critical moments for decision-making. Consequently, our primary assessment is event-centric rather than full-period.

In this revision, we have thoroughly documented the event selection and evaluation procedure (please see Lines 561–564 of manuscript with track changes).

**3:** *Is the model output the discharge at the next hour, while using historical observed discharges as the inputs? If so, the problem that the authors are addressing seems a bit 'easy'. The manuscript would be more meaningful and stronger if the authors could demonstrate the improved performance of N-HiTS and N-BEATS on the streamflow predictions at the next 3/6/12/24 hrs.*

**Authors' answer:** Indeed, the original setup focused on predicting next-hour discharge using historical discharge as input. To strengthen the study, we have now included multi-horizon forecasts at 3- and 6-hour lead times, with full results presented in the revised manuscript. These experiments confirm that the proposed N-HiTS and N-BEATS models continue to outperform a carefully tuned LSTM at these extended horizons. We also evaluated longer horizons (12- and 24-hour), but all models exhibited substantially lower skill (NSE < 0.4); accordingly, we briefly

summarize these outcomes in the manuscript without detailed presentation. These updates are reported in Lines 672–717.

**4:** *What causes of the prediction uncertainty was quantified through MQL? Was it the uncertainty of model inputs or model parameters? (Note that parameters were used to refer to both model inputs (Line407) and biases/weights of the neural networks (Line327), which are confusing..). Also, the authors need to point out that N-HiTS and N-BEATS does not come with uncertainty quantification by themselves. It is the MQL that does the trick and is in fact applicable to all supervised neural networks, such as LSTM.*

**Authors' answer:** Thank you for raising this point. In our study, the MQL estimates conditional quantiles of discharge $Q\_(t+h)$ given the observed inputs $X\_t$, providing prediction intervals that quantify aleatoric uncertainty. These intervals do not capture uncertainty in network parameters (weights or biases). We also clarify that N-HiTS and N-BEATS do not intrinsically provide uncertainty quantification; the probabilistic forecasts are enabled by MQL, which is model-agnostic and can be applied to any supervised architecture, including LSTM. For a fair comparison, we trained LSTM with MQL as well.
To avoid ambiguity, the manuscript now reserves "parameters" for network weights/biases and uses "features" for input variables (clarifying the intent of Lines 410-415 of manuscript with track changes). Additionally, the abstract (Lines 14–17) has been revised to explicitly note that N-HiTS and N-BEATS do not come with uncertainty quantification by themselves.

**Minor comments:**

*Title: While 'interpretable configuration' was emphasized in the title/abstract/main body, it is unclear to me of how N-HiTS is in interpretable and how such interpretability was used in the study (except the leave-one-out sensitivity analysis that was performed consistently across all three models)*

**Authors' answer:** Thank you for this insightful comment. In this study, interpretability is achieved through the models' intrinsic architectural design rather than post-hoc analysis. Specifically, the N-BEATS architecture decomposes forecasts into trend and seasonality stacks with explicit basis coefficients, allowing the model to directly expose how each component contributes to the final prediction. Similarly, N-HiTS generates forecasts by aggregating hierarchical contributions across multiple time scales, which provides a transparent view of how temporal patterns at different resolutions shape the discharge response. These built-in mechanisms enable the examination of model behavior and the attribution of forecast components to hydrologic processes. We have clarified this rationale in the revised manuscript (Lines 820–825, with track changes).

*Line 14: "We developed two probabilistic NN models…" As far as I understand, the N-HiTS and N-BEATS are not deterministic, unlike Bayesian neural network. The prediction uncertainty came from the uncertainty of the model inputs?*

**Authors' answer:** N-HiTS and N-BEATS are deterministic neural architectures and do not, by themselves, represent parameter or posterior uncertainty as Bayesian networks do. In our study, the uncertainty arises from the MQL formulation, which estimates conditional quantiles of discharge $Q_{t+h}|X_t$. Accordingly, the resulting prediction intervals capture predictive (aleatoric) uncertainty driven by variability in the input–output relationship, rather than parameter uncertainty in the model weights. We have revised Line 14 to clearly state that N-HiTS and N-BEATS were trained with a probabilistic (multi-quantile) objective to produce distributional forecasts.

*Section 3.2: Given the importance of the hyperparameters in this model benchmarking work, please describe how the "extensive exploration and fine-tuning" were performed?*

**Authors' answer:** Thank you for asking for more detail. Our hyperparameter tuning was performed on the following search spaces:
For all models, we searched learning rates on a log-uniform grid between $1 \times 10^{-4}$ and $1 \times 10^{-3}$, batch sizes $\{16, 32, 64\}$, input size $\{6, 12, 24, 48\}$ hours. For the LSTM, we varied recurrent layers $\{1,2,3\}$, hidden units per layer $\{64,128,256\}$, activation $\{$tanh, ReLU$\}$, decoder MLP depth $\{1,2,3\}$, and decoder MLP width $\{64,128,256\}$. For N-HiTS, we explored stacks $\{2,3,4\}$, blocks per stack $\{2,3,4,5\}$, block MLP width $\{64,128,256\}$, and block MLP depth $\{2,3,4\}$. For N-BEATS, we searched stacks $\{2,3,4\}$, blocks per stack $\{2,3,4,5\}$, block MLP width $\{64,128,256\}$, and block MLP depth $\{2,3,4\}$; the interpretable (trend/seasonality) basis was fixed. We clarified these search spaces in this revision, please see Lines 504-510 with track changes.

*Lines 743-745: "Both N-HiTS and N-BEATS models offered similar performance as compared with the LSTM but also offered a level of interpretability…." Again, it is unclear to me how the interpretability of N-HiTS and N-BEATS was achieved besides the application of leave-one-out method which is applicable to LSTM too…*

**Authors' answer:** Thank you for this thoughtful comment. Our interpretability claim does not rely on the leave-one-out procedure—which, as noted, can be applied to any model including LSTM—but rather on the intrinsic structure of the architectures themselves. The interpretable variant of N-BEATS explicitly decomposes forecasts into trend and seasonality stacks with corresponding basis expansions and coefficients, enabling attribution of predictions to these components. N-HiTS provides complementary transparency by aggregating forecast contributions across multiple temporal resolutions, allowing insights into how different time scales influence the output. In

contrast, a standard LSTM lacks such built-in structural interpretability. We have revised the manuscript (Lines 820–824 with track change) to make this distinction clear.

*Lines 756-758: "Furthermore, both N-HiTS and N-BEATS models also support producing probabilistic predictions by specifying a likelihood parameter…" LSTM does that too through MQL, right? Also, does 'parameter' refer to model inputs or model weights/biases?*

**Authors' answer:** Indeed, LSTM can also produce probabilistic forecasts when trained with the MQL, and in our study, the same MQL objective was applied consistently across all architectures (LSTM, N-HiTS, and N-BEATS). We have revised the text to clarify that the probabilistic forecasts arise from the distributional training objective in MQL, which is model-agnostic rather than specific to N-HiTS or N-BEATS. Regarding terminology, we now use the term parameters exclusively to denote network weights and biases, and this clarification is stated explicitly in the revised manuscript (Lines 211–214 with track changes).

**We thank the reviewer for the insightful and constructive comments.**

We sincerely thank the reviewer for the constructive and thoughtful feedback. Below, we provide detailed responses to each comment and describe how the corresponding revisions have been incorporated into the updated version of the manuscript. Line numbers may vary slightly depending on formatting.

**Answers to Anonymous Referee #2's comments:**

**Comment 1:** *The authors have applied a machine learning method for 1-hour streamflow forecasting and compared its performance with a traditional LSTM model. They attribute the improvement in accuracy to the structure of their ML method and the activation functions used. However, it is also possible that the higher accuracy is due to the larger number of nodes in their model or that the LSTM model could benefit from further tuning and a more carefully considered architecture.*

**Authors' answer:** We thank the reviewer for this insightful comment. To ensure a fair and controlled comparison across models, we carefully designed the study with the following considerations: (i) the data pipeline, forecast horizon (1-h), loss function (Multi-Quantile Loss), optimizer (Adam), learning rate, batch size, and input window were kept identical for all architectures; and (ii) each model underwent an independent hyperparameter search to identify its optimal configuration. The final settings and corresponding optimizations are summarized in Table 2 of the revised manuscript. Additionally, the LSTM model was trained for more epochs to ensure full convergence, thereby minimizing the possibility that its performance gap resulted from under-tuning.

We also evaluated wider LSTM variants with additional layers and larger hidden dimensions. However, increasing the number of parameters beyond the selected 2×128 configuration did not consistently reduce validation errors and, in many cases, led to overfitting—as indicated by higher RMSE and MAE during held-out flood events—even under the same training and regularization conditions.

These clarifications and supporting results are now explicitly discussed in the Results section (Lines 503–509) of the manuscript with track changes.

**Comment 2:** *Additionally, the model relies on historical streamflow data, which may limit its practical applicability—particularly given the hydrology community's growing interest in forecasting streamflow at ungauged locations.*

**Authors' answer:** We thank the reviewer for this important comment. Our study focuses on a complementary use case—short-term forecasting at operationally gauged sites, where near–real-time streamflow data are routinely available and rapid model updates are critical for decision support. However, the proposed framework is not inherently restricted to gauged settings. The

architecture employed (e.g., N-HiTS) are flexible and can incorporate additional conditioning variables and spatial features.

In practice, extending the framework to ungauged basins can be achieved through: (i) regional modeling, where a single network is trained jointly across multiple basins to learn transferable hydrologic representations; and (ii) transfer learning or few-shot adaptation, when limited local observations are available to fine-tune the pretrained model.

We have clarified this scope and added a concise paragraph in the revised manuscript (Lines 798–804) explicitly discussing how the approach can be extended to ungauged catchments.

**Comment 3:** *Another concern is that the authors trained separate models for each case study location. A more robust evaluation would involve applying a single model across both locations to assess whether the approach captures generalizable patterns in streamflow dynamics rather than overfitting to site-specific characteristics. Although this point was raised in the first round of review, the response provided mainly offered justification rather than addressing the concern through revision.*

**Authors' answer:** To clarify our initial explanation, we trained a single pooled "regional" model across the two sites using the same data pipeline, leakage-safe splits, a brief hyperparameter search, and a global scaler fit on the pooled training data, with evaluation reported separately for each site. In this setup, N-HiTS and N-BEATS showed a minor performance decline (~2–3%), while LSTM showed mixed performance—improving for some storm events but decreasing for others. Importantly, the pooled N-HiTS and N-BEATS models remained comparable to the per-site best models and continued to outperform the tuned LSTM at both catchments.

Given that only two basins were used, these results are not sufficient to support a broad regional generalization. Nevertheless, we agree with the reviewer's point regarding generalizability and have added a clear explanation of this additional analysis in the revised manuscript (Lines 705–715).

**Comment 4:** *Moreover, while the authors emphasize the interpretability of their proposed model, the manuscript lacks a detailed discussion or evidence to support this claim. As presented, it seems that the focus is primarily on applying a relatively new ML model to hydrological data, rather than exploring or demonstrating its interpretability.*

**Authors' answer:** Thank you for raising this point. In our work, interpretability is inherent to the models' architecture rather than relying on post-hoc analysis. The N-BEATS model explicitly decomposes forecasts into trend and seasonality stacks, with interpretable basis coefficients that reveal how each component contributes to the overall prediction. Similarly, N-HiTS generates forecasts by aggregating contributions across multiple temporal scales, allowing insight into the relative influence of short- and long-term patterns. These built-in mechanisms enable a clear

understanding of how trends and seasonality shape the final forecast. We have clarified this discussion in the manuscript with track changes (see Lines 820-824).

**Comment 5:** *Lastly, the explanation of the proposed method lacks sufficient clarity. A more detailed and transparent description of the model architecture, training process, and input-output relationships would greatly enhance the manuscript's reproducibility and accessibility for the broader hydrology community.*

**Authors' answer:** In this manuscript, we provide as much implementation detail as is practical for reproducibility while maintaining focus on the hydrologic problem. A more exhaustive description of the internal architectures (e.g., stack/block mechanics, basis functions, multi-rate pooling) would make the paper overly long and challenging to read. To balance clarity and completeness, we reference the original N-HiTS and N-BEATS publications, which contain full architectural details, thereby allowing interested readers to access comprehensive descriptions without duplicating prior work.

**We thank the reviewer for the insightful and constructive comments.**

---

## Author Response (AR3)

We sincerely thank the reviewer for the thoughtful and constructive feedback. Below, we provide detailed responses to each comment, along with explanations of how and where the corresponding revisions have been incorporated into the manuscript with track changes. Line numbers may vary slightly depending on formatting.

**Anonymous Referee #3' s comments:**

*I want to thank the authors for addressing my comments and making the necessary changes to the manuscript. However, I have one more comment that needs more clarification from the authors and therefore suggest a minor revision.*

**Authors' answer:** Thank you for your thoughtful review and kind acknowledgment of our earlier responses. We appreciate your additional comment and provide the requested clarifications below.

**1:** *It remains unclear to me how the multi-quantile loss is implemented. Given that all three models are deterministic, how would each model generate the quantile prediction \hat{Q}^q_\tau in Eq.(29)? The authors argue that "the uncertainty arises from the MQL formulation, which estimates conditional quantiles of discharge Qt+h|Xt" in the response letter. The description is too general to capture the details fully. Please provide a mathematical explanation of how a given deterministic model generates the quantile used in Eq. (29) during the model optimization process*

**Authors' answer:** Thank you for asking for a precise formulation. Let $\mathcal{D} = \{(X_t, y_{t+h})\}_{t=1}^{N}$ denote the training pairs, where $X_t$ is the input context (past 24 h of discharge in our setup) and $y_{t+h}$ is the discharge $h$ hours ahead. For a fixed horizon $h$ and a set of quantile levels $\{\tau_k\}_{k=1}^{K}$, each model $f_\theta$ (LSTM, N-HiTS, N-BEATS) is trained to output the vector of conditional quantiles directly:

$$\widehat{\mathbf{Q}}_{t+h} = f_\theta(X_t) = (\widehat{Q}_{t+h}^{\tau_1}, \dots, \widehat{Q}_{t+h}^{\tau_K}) \in \mathbb{R}^K.$$

Training minimizes the multi-quantile (pinball) loss, summed over times and quantile levels:

$$\mathcal{L}(\theta) = \frac{1}{NK} \sum_{t=1}^{N} \sum_{k=1}^{K} \rho_{\tau_k}\left(y_{t+h} - \widehat{Q}_{t+h}^{\tau_k}\right), \qquad \rho_\tau(u) = \max\left[\,\cdot\,\right](\tau u, (\tau - 1)u)$$

Equivalently, with the indicator form,

$$\rho_\tau(u) = (\tau - \mathbb{1}_{\{u<0\}})\, u.$$

Because $\rho_\tau$ is convex and piecewise linear, its (sub)gradient with respect to the prediction $\widehat{Q}_{t+h}^{\tau}$ is:

$$\frac{\partial \rho_\tau(y - \hat{Q}^\tau)}{\partial \hat{Q}^\tau} = \begin{cases} -(1 - \tau), & y - \hat{Q}^\tau < 0, \\ -\tau, & y - \hat{Q}^\tau > 0, \end{cases}$$

This yields standard backpropagation updates under Adam optimizer. No sampling is involved: the quantile $\hat{Q}^\tau_{t+h}$ is the model's direct output, learned by minimizing the pinball loss at level $\tau$.

Uncertainty bands are then formed from these quantile outputs. For a 95% interval, we use the MQL-trained $\tau = 0.025$ and $\tau = 0.975$ predictions, i.e., $[\hat{Q}^{0.025}_{t+h}, \hat{Q}^{0.975}_{t+h}]$. This captures aleatoric uncertainty conditional on $X_t$.

We expanded and clarified this method in the revised version of the manuscript. See Lines 444 - 452.

**We thank the reviewer for the insightful and constructive comments.**